# Highly Pathogenic Avian Influenza: Tracking the Progression from IAV (H5N1) to IAV (H7N9) and Preparing for Emerging Challenges

**DOI:** 10.3390/microorganisms14010012

**Published:** 2025-12-19

**Authors:** Mahmoud H. El-Bidawy, Imran Mohammad, Md. Rizwan Ansari, Mohammed Ibrahim Hajelbashir, Mohammed Sarosh Khan, Muhammad Musthafa Poyil, Md. Nadeem Bari, Abdullah M. R. Arafah, Mohammad Azhar Kamal, Shaheena Tabassum Mohammad Ahsan

**Affiliations:** 1Department of Basic Medical Sciences, College of Medicine, Prince Sattam Bin Abdulaziz University, Al-Kharj 11942, Saudi Arabiai.mohammad@psau.edu.sa (I.M.);; 2Department of Pediatric, College of Medicine, Prince Sattam Bin Abdulaziz University, Al-Kharj 11942, Saudi Arabia; 3Department of Pharmaceutics, College of Pharmacy, Prince Sattam Bin Abdulaziz University, Al-Kharj 11942, Saudi Arabia

**Keywords:** highly pathogenic avian influenza (HPAI), IAV (H5N1), IAV (H7N9), zoonotic transmission, One Health approach, viral evolution, global surveillance

## Abstract

Highly Pathogenic Avian Influenza (HPAI) viruses, particularly IAV (H5N1), continue to pose a major global threat due to their widespread circulation and high mortality rates in birds. Management of HPAI is complicated by challenges in conserving migratory bird populations, sustaining poultry production, and uncertainties in disease dynamics. Structured decision-making frameworks, such as those based on the PrOACT model, are recommended to improve outbreak response and guide critical actions, especially when HPAI virus (HPAIV) detections occur in sensitive areas like wildlife refuges. Surveillance data from late 2024 to early 2025 show persistent HPAI activity, with 743 detections across 22 European countries and beyond, and notable outbreaks in poultry in nations like Hungary, Iceland, and the UK. The proximity of poultry farms to water sources increases environmental contamination risks. Meanwhile, HPAI A(IAV (H5N1)) and other H5Nx viruses have been detected in a wide range of mammalian species globally, raising concerns about mammalian adaptation due to mutations like E627K and D701N in the PB2 protein. Human infections with IAV (H5N1) have also been reported, with recent cases in North America highlighting zoonotic transmission risks. Molecular studies emphasize the importance of monitoring genetic variations associated with increased virulence and antiviral resistance. Preventive strategies focus on biosafety, personal protective measures, and vaccine development for both avian and human populations. Ongoing genetic characterization and vigilant surveillance remain critical to managing the evolving threat posed by HPAI viruses.

## 1. Introduction

### 1.1. Background on Avian Influenza Viruses

Avian influenza viruses (AIVs) are members of the *Orthomyxoviridae* family and possess a segmented, negative-sense RNA genome encoding at least 14 proteins. According to the Baltimore classification system, IAVs are classified in Group V, which includes viruses with negative-sense, single-stranded RNA genomes, classified under the genus *Alphainfluenzavirus* (formerly known as *Influenza A Virus*). These enveloped viruses possess a segmented RNA genome consisting of eight gene segments and are capable of infecting a wide range of vertebrate hosts, primarily avian species but occasionally mammals and humans. IAVs are characterized by combinations of two surface glycoproteins: hemagglutinin (HA) and neuraminidase (NA) [1,2]. In birds, 16 HA and 9 NA antigens have been identified, enabling the classification of numerous subtypes (e.g., IAV (H5N1), H7N8). To date, 19 HA (H1–H19) and 11 NA (N1–N11) subtypes have been recognized. Of these, H1–H16 and H19 have been detected in wild birds and 11 NA (N1–N11) subtypes have been recognized, with H17N10 and H18N11 isolated exclusively from bats [3,4].

Based on their virulence in chickens, AIVs are further categorized into two groups: low-pathogenic avian influenza viruses (LPAIVs) and highly pathogenic *avian influenza viruses* (HPAIVs) [1]. LPAIVs typically induce mild or asymptomatic infections, occasionally causing respiratory symptoms, weight loss, or decreased egg production in poultry [1]. In contrast, HPAIVs, historically referred to as “fowl plague,” are associated with severe systemic disease and high mortality rates. To date, only viruses of the H5 and H7 subtypes have been shown to evolve into highly pathogenic forms [3].

Among these, HPAI (H5N1) has emerged as a particularly concerning zoonotic and pandemic threat, demonstrating an expanded host range, enhanced tissue tropism, and increased virulence. The first major zoonotic outbreak of HPAI (H5N1) occurred during an epizootic in Hong Kong in 1997 (Table 1), resulting in 18 human infections and 6 fatalities. Although an earlier outbreak of HPAI (H5N1) among poultry was recorded in Scotland in 1959, the 1997 Hong Kong event marked the first known human infections [5]. Following a brief period of quiescence, HPAI (H5N1) re-emerged in 2003, subsequently establishing a global presence with incursions into East Asia, Southeast Asia, West Asia, and Africa between 2003 and 2008 [6].

Although both the 1959 and 1997 HPAI (H5N1) outbreaks involved highly pathogenic H5N1 viruses, they originated from distinct evolutionary lineages. The 1959 outbreak in Scotland was caused by a virus belonging to the Eurasian H5 lineage, which was not part of the Goose/Guangdong (Gs/Gd) lineage and has not been assigned to the modern clade classification system [7]. In contrast, the 1997 Hong Kong outbreak, which marked the first known zoonotic transmission of H5N1 to humans, originated from the A/goose/Guangdong/1/1996 (Gs/Gd) lineage. This Gs/Gd lineage is the sole H5N1 lineage that has undergone extensive diversification into multiple clades and subclades (e.g., clades 0, 1, 2.2, 2.3.4.4b), as referenced in Table 1 and global surveillance frameworks (WHO/WOAH). All subsequent H5N1 panzootics, including the currently circulating clade 2.3.4.4b viruses, are descendants of this Gs/Gd progenitor [7].

**Table 1 microorganisms-14-00012-t001:** Global distribution of IAV (H5N1) and IAV (H7N9) outbreaks.

Virus Subtype	Timeline/Year(s)	Region/Country	Major Events and Notes
IAV (H5N1)	1997	Hong Kong (SAR)	First human outbreak (Clade 0) [8]
	2003 onwards	SE Asia (Vietnam, Laos, Thailand), China	Widespread outbreaks (Clade 1) [8]
	2005–2006	Asia, Europe, Africa	Intercontinental spread (Clade 2.2) [8]
	2014–2015	China, Korea, USA, Europe	H5N8 spread (Clade 2.3.4.4a) [8,9]
	2016–2017	Global	H5Ny emergence (Clade 2.3.4.4b) [8,10,11]
	2020–2021	Europe, Asia, Africa	Novel IAV (H5N1) clade 2.3.4.4b wave [10,11]
	Late 2021–ongoing	North America	Mass poultry/mammal outbreaks [8,10]
	Late 2022–ongoing	Mexico, Central and South America	Spread of clade 2.3.4.4b [8,10]
	2023	Europe (Poland, Spain, Finland)	Infections in cats, mammals [8,11]
	2024–ongoing	USA (Dairy cattle), Global	Dairy cattle infection, continued wild bird circulation [10]
IAV (H7N9)	2019	Cambodia	Emergence of H7N4 [12,13]
	2003	Netherlands	IAV (H7N7) outbreak (human infections) [12]
	2013–2017	China	Five epidemic waves, LPM interventions successful [14]
IAV (H9N2)	1998–ongoing	China, Egypt, Pakistan, Oman	Endemic poultry infection; mild human disease [13]

More recently, a Eurasian-origin IAV (H5N1) virus belonging to clade 2.3.4.4b was introduced into North America via migratory birds in late 2021 [10,11], resulting in an unprecedented epizootic affecting poultry, wild birds, and numerous mammalian species (Figure 1). By late 2022, the outbreak expanded into Mexico, Central America, and South America. Concurrently, between 2020 and 2021, Europe, Asia, and Africa witnessed a novel wave of clade 2.3.4.4b IAV (H5N1) activity, severely impacting wild and domestic avian populations [8].

Distinctly, a novel reasserting IAV (H7N9) virus emerged in March 2013 in China’s Shanghai and Anhui provinces, linked to live poultry markets. Genetic analyses revealed that IAV (H7N9) arose through multiple assortment events involving wild bird viruses, with internal genes derived from endemic IAV (H9N2) viruses and external HA and NA segments from separate lineages [13]. Domestic chickens subsequently became the primary reservoir for IAV (H7N9) [8]. Despite being classified as low pathogenic in birds, IAV (H7N9) caused 131 confirmed human infections and 36 deaths within two months of its emergence, raising global concern. The IAV (H7N9) epidemic in China evolved through five distinct waves between 2013 and 2017 (Table 1), culminating in the emergence of highly pathogenic variants during the fifth wave. Although large-scale outbreaks have since diminished, IAV (H7N9) viruses remain a latent threat to global public health [15].

In addition to IAV (H7N9), other avian influenza subtypes, including IAV (H7N3), IAV (H5N8), and IAV (H5N5), have also played significant roles in recent epizootics. On the H7 side, while H7N9 remains the most concerning for zoonotic risk, H7N3 has caused repeated outbreaks in poultry in the Americas and led to sporadic human infections (e.g., Mexico, 2012 and 2016), underscoring its pandemic potential. Regarding H5 viruses, highly pathogenic H5N8 (clade 2.3.4.4b)—first detected in poultry in Asia in 2014—rapidly spread globally via migratory birds, causing massive poultry losses across Europe, Africa, and the Americas. Similarly, H5N5 viruses, often reassortants with H5N8, have emerged in wild birds and poultry in Europe and North America, contributing to ongoing panzootic circulation [8,16].

And also Evolutionary analysis revealed that the hemagglutinin (HA) gene of the novel IAV (H5N9) virus was derived from A/Muscovy duck/Vietnam/LBM227/2012 IAV (H5N1), a strain belonging to clade 2.3.2.1, while its neuraminidase (NA) gene originated from the human-infecting strain A/Hangzhou/1/2013, IAV (H7N9) [17].

### 1.2. Virology and Molecular Determinants of Pathogenicity

*Influenza A viruses* (IAVs) belong to the *Orthomyxoviridae* family and possess a segmented, negative-sense RNA genome encoding at least 14 proteins, including the surface glycoproteins hemagglutinin (HA) and neuraminidase (NA)—key mediators of viral entry and release, respectively [18]. The interplay between HA, NA, and host cell receptors governs viral tropism, transmission efficiency, and disease severity.

#### 1.2.1. Receptor Specificity and Host Tropism

AIVs primarily recognize sialic acid (SA) residues on host cell surface glycoconjugates as entry receptors. The linkage type of SA—specifically α2,3-linked versus α2,6-linked—is a major determinant of host range. Avian species predominantly express α2,3-SA in their intestinal and respiratory tracts [19], which is preferentially bound by most avian-origin HA subtypes (e.g., H5, H7). In contrast, human upper respiratory epithelia are enriched in α2,6-SA, typically recognized by human-adapted influenza viruses [10,11].

This receptor specificity explains why zoonotic IAV (H5N1) infections in humans often result in lower respiratory tract involvement (where α2,3-SA is still present), leading to severe pneumonia rather than efficient human-to-human transmission [20,21]. Notably, ocular conjunctival epithelial cells also express α2,3-SA, providing a plausible route for IAV (H5N1) entry via the eye—consistent with reported cases of IAV (H5N1)-associated conjunctivitis [10].

#### 1.2.2. HA: Mediator of Viral Entry and a Key Virulence Factor

HA facilitates viral attachment to SA receptors and subsequent fusion of the viral and endosomal membranes following endocytosis. The cleavability of HA by host proteases is a critical determinant of systemic spread and pathogenicity. Low-pathogenicity AIVs (LPAIVs) possess a single basic amino acid at the HA cleavage site [1,3], restricting cleavage to trypsin-like proteases found in the respiratory or intestinal tract. In contrast, highly pathogenic AIVs (HPAIVs), such as contemporary IAV (H5N1) clade 2.3.4.4b viruses, harbor a polybasic HA cleavage site (e.g., RRRKR↓), enabling cleavage by ubiquitous furin-like proteases—leading to systemic infection and multi-organ failure [1,10,22].

Recent studies (2024–2025) confirm that mutations enhancing HA stability at higher pH or improving binding affinity to α2,3-SA further increase virulence in mammalian models [11].

#### 1.2.3. NA: Enabling Viral Release and Balancing HA Binding

Following replication, newly assembled virions remain tethered to the host cell surface via HA–SA interactions. Neuraminidase (NA) cleaves terminal SA residues, facilitating the release and dispersal of progeny virions and preventing self-aggregation [2,23]. An optimal functional balance between HA binding avidity and NA enzymatic activity is essential for efficient viral replication and transmission [2,19].

Disruption of this balance—e.g., through NA inhibitor resistance mutations or HA mutations that increase receptor affinity without compensatory NA adaptations—can impair viral fitness. However, certain NA stalk deletions common in IAV (H5N1) clades) may enhance replication in poultry by modulating this HA–NA interplay [3,13].

### 1.3. Significance of the Topic

For avian influenza viruses to cross the species barrier and pose a zoonotic threat, they must overcome key host-range restrictions. This adaptation typically requires mutations in the viral genome that alter receptor-binding preference from avian-type (α2,3-linked sialic acid) to human-type (α2,6-linked sialic acid) receptors and enhance replication efficiency in mammalian cells. The specific mutations that confer this adaptive potential, particularly in key proteins like hemagglutinin (HA) and the polymerase subunit PB2, vary between virus subtypes and are a critical focus of evolutionary surveillance. The detailed adaptation mechanisms for IAV (H5N1) and IAV (H7N9) are discussed in Section 3.2 [24]. AIV, particularly HPAI (H5N1) and IAV (H7N9), continue to pose substantial threats to global health due to their extensive circulation, high mortality rates (Figure 2), and pandemic potential. Persistent viral circulation among avian populations serves as a reservoir, facilitating spillover into mammals, including species of agricultural and economic importance, as well as humans.

Recent epizootics, notably the HPAI (H5N1) outbreaks involving mammals such as sea lions and dairy cattle, underscore the urgent need for comprehensive virological surveillance and intersectoral response strategies. The growing number of human infections signals an increasing risk of viral adaptation toward more efficient human-to-human transmission. Although direct avian-to-human transmission remains relatively rare, the expanding host range and unprecedented scale of current outbreaks heighten concerns about viral evolution and persistence across species barriers [12].

Of particular alarm is the emergence of HPAI (H5N1) infections in dairy cattle. Should cattle become long-term reservoirs, they could facilitate further viral reassortment and adaptation, significantly escalating the risk of zoonotic transmission (The virus has zoonotic potential (to humans) and a broad cross-species transmission potential among mammals). Additionally, cross-species transmission among mammals raises the specter of viral mutations favoring sustained mammalian or human transmission [25].

Similarly, IAV (H7N9) remains classified as a high-risk virus for pandemic potential. Should either H5 or H7 subtype viruses acquire mutations enabling efficient aerosolized transmission among humans, they could precipitate a pandemic with catastrophic public health and socioeconomic consequences [26].

Understanding the epidemiological dynamics, molecular evolution, and zoonotic potential of IAV (H5N1) and IAV (H7N9) is therefore critical for enhancing pandemic preparedness, informing vaccination strategies, and implementing effective public health interventions. The ongoing global outbreak of avian influenza highlights the fragile balance between human activity and viral ecology, demanding vigilant international cooperation [27].

This review complements recently published Review article by Possas et al. (2025) [28], which emphasized vaccine governance and artificial-intelligence-driven preparedness strategies. In contrast, the present paper provides a comparative, data-driven synthesis of IAV (H5N1) and IAV (H7N9), integrating updated epidemiological data through March 2025, detailed genomic mutation mapping, clinical phenotype comparison, and practical One-Health recommendations aimed at operational response and policy planning.

## 2. Epidemiology of IAV (H5N1) and IAV (H7N9)

### 2.1. Global Distribution and Spread

IAV (H5N1) is now recognized as a panzootic virus, with sustained transmission across multiple continents. It continues to circulate widely among wild birds, domestic poultry, and an increasing number of mammalian species, indicating its successful ecological establishment and global persistence [10,11,18].

#### 2.1.1. IAV (H5N1)

Highly pathogenic Avian influenza virus (HPAIV) IAV (H5N1) first gained international attention following an epizootic event in Hong Kong in 1997 [22], where it caused 18 human infections and six deaths (Table 2). Although the initial detection of HPAI (H5N1) in poultry was documented earlier in Scotland in 1959, the 1997 outbreak marked the virus’s first recognized zoonotic transmission to humans [5].

After a temporary hiatus, HPAI (H5N1) re-emerged in 2003 and has since become enzootic in several regions, particularly across Europe, Asia, and Africa. Between 2003 and 2008, the virus expanded its geographic footprint to East Asia, Southeast Asia, West Asia, and North Africa. While cases remained relatively low across most of Asia between 2013 and 2015, Egypt experienced a marked surge during 2014–2015 [29].

**Table 2 microorganisms-14-00012-t002:** Cumulative Number of Human Cases and Mortality for IAV (H5N1), IAV (H7N9), and IAV (H9N2), WHO [30].

Virus Subtype	Year Range	Total Cases	Fatalities	Case Fatality Rate (%)	Notes
IAV (H5N1)	1997	18	6	33.3%	First human cases (Hong Kong) [31,32]
	2003	3	2	66.7%	Re-emergence in Asia [31,32]
	2004	46	32	69.6%	Vietnam, Thailand, others [31,32]
	2003–14 July 2023	878	458	52.16%	WHO cumulative reports [30]
	March–October 2024 (USA)	4	1	25%	Exposure to cows and poultry [30,33,34]
	Since 1997–11 December 2024	974	464	47.6%	WHO cumulative reports [30]
	Since 1997–7 March 2025	989	467	47.2%	WHO Updated data [30]
IAV (H7N9)	March–May 2013 (China)	130	≥27	20%	First epidemic wave in China [31,32,35]
	Since 2013 (China, five waves)	1567	615	39.2%	Five epidemic waves [31,32,35]
IAV (H9N2)	1998–2019	59	1	1.7%	Mild infections [13,31,32]

Notes: IAV (H9N2) is placed in this table just for comparative purposes as it contributes to re-assortment and zoonotic infections, although classified as LPAI. Case Fatality Rate (CFR) calculated as (Fatalities/Total Cases) × 100 based on the data presented.

A major shift occurred between 2020 and 2021, with the emergence of HPAI (H5N1) clade 2.3.4.4b across Europe, Asia, and Africa. This lineage caused widespread outbreaks among wild and domestic birds, resulting in the largest recorded HPAI epidemic during the 2021–2022 season. Over 2000 outbreaks were reported across 37 European countries, necessitating the culling of more than 40 million birds [3].

In late 2021, Eurasian-origin IAV (H5N1) clade 2.3.4.4b entered North America via migratory birds, igniting an unprecedented epizootic in poultry, wild birds, and mammals (Figure 1). From 2022 onwards, the virus expanded further into Mexico, Central America, and South America [25]. Between January 2022 and June 2023, IAV (H5N1) infections were confirmed in over 7000 wild birds across all 50 U.S. states and over 800 poultry flocks across 47 states—an unprecedented geographic spread not seen since 2016 [5,36].

Today, clade 2.3.4.4b IAV (H5N1) viruses are endemic in many regions and have been detected on every continent except Australia, extending even into urban areas and Antarctica. The ancestral lineage of these viruses can be traced back to A/goose/Guangdong/1/96, from which multiple genetic clades have evolved [3].

#### 2.1.2. IAV (H7N9)

In March 2013, a novel reassortant IAV (H7N9)) virus was first detected in Shanghai and Anhui provinces of China. Genetic analyses revealed that IAV (H7N9) originated through multiple reassortment events in wild birds, acquiring internal genes from circulating IAV (H9N2) viruses and external HA and NA segments from separate sources. The virus subsequently adapted to domestic poultry, particularly chickens, which became its primary reservoir [37].

Although low pathogenic in birds, IAV (H7N9) caused 131 human infections and 36 fatalities within just two months of its emergence (Table 2), raising considerable international concern. Between 2013 and 2017, IAV (H7N9) outbreaks occurred in five epidemic waves [8], with each wave beginning in the winter (October–November) and subsiding by early summer (June). The 2016–2017 wave was the most severe, accounting for 759 confirmed human cases due to a broader geographical spread across China [38].

During the fifth wave, a highly pathogenic variant of IAV (H7N9) HPIAV (H7N9) emerged, increasing the risk to both poultry and humans. Two genetically distinct lineages—the Pearl River Delta and Yangtze River Delta lineages—have since been identified. Although large-scale epidemics have subsided, sporadic poultry outbreaks continue, and IAV (H7N9) remains a latent public health threat [14].

### 2.2. Case Fatality Rates and Clinical Outcomes

#### 2.2.1. IAV (H5N1)

Between 2003 and July 2023, the World Health Organization (WHO) documented 868 confirmed human cases of HPAI (H5N1) infection across 23 countries, resulting in 458 fatalities, a case fatality rate (CFR) of approximately 52.2%. The majority of these infections occurred prior to 2016. From 2022 to July 2023 alone [5], 14 cases and two fatalities were reported (Table 2).

Historically, the CFR for IAV (H5N1) infections has been reported at around 60%, substantially higher than that of pandemic influenza viruses [8]. Clinical outcomes are often severe, characterized primarily by viral pneumonia and progression to acute respiratory distress syndrome (ARDS). The pathogenesis of HPAI (H5N1) in humans is complex and multifactorial, involving direct viral cytopathic effects and dysregulated host immune responses [39].

#### 2.2.2. IAV (H7N9)

The initial outbreak of IAV (H7N9) in 2013 led to 131 confirmed human cases and 36 deaths within a short timeframe. As of May 2013, 130 confirmed cases (excluding mainland China) had been reported, with at least 27 deaths [38], indicating a mortality rate of approximately 20% (Table 2).

Across five epidemic waves from 2013 to 2017, H7N9 caused 1567 confirmed human 255 infections and 615 deaths, corresponding to a cumulative CFR of approximately 39.2%. 256 Severe disease manifestations primarily include viral pneumonia and ARDS. Unlike 257 H5N1, H7N9 infections are often associated with underlying comorbidities [14]. e.g., H5N1 clade 2.3.4.4b, H7N9, and H5N6) associated with high human case fatality rates (30–60%) as reported by WHO and CDC data [35,38]. Unlike H5N1, H7N9 infections are often associated with underlying comorbidities—particularly hypertension, diabetes mellitus, chronic respiratory diseases, and cardiovascular conditions—which have been consistently linked to increased risk of severe outcomes and higher mortality [14,15].

Low-pathogenic IAV (H7N9) viruses were predominant during the first four waves, while highly pathogenic strains emerged in the fifth wave. Although both LPIAV (H7N9) and HPIAV (H7N9) viruses infected humans, current evidence does not suggest a significant difference in virulence or transmissibility between these two variants in human hosts [14,15,20].

### 2.3. Zoonotic Potential

Role of Poultry Farming, Live Bird Markets, and Migratory Birds

AIVs, particularly subtypes H1 through H16, are naturally maintained in wild aquatic bird populations, which serve as their primary reservoir. Traditionally, wild waterfowl carried AIVs subclinical. However, since 2005, HPAI (H5N1) has increasingly caused morbidity and mortality among wild birds, altering the epidemiological landscape [21].

IAV (H5N1) viruses, especially clade 2.3.4.4b, have been frequently detected in wild birds and have caused extensive die-offs, including among endangered species. Transmission from wild birds to domestic poultry occurs primarily via the fecal–oral route, direct contact, and, to a lesser extent, aerosol transmission [5,21].

Migratory birds have played a critical role in disseminating HPAI viruses across continents. Notably, Eurasian-origin clade 2.3.4.4b viruses were introduced into North America in late 2021 via trans-Atlantic migratory pathways, subsequently spreading across the Americas. Wild bird infections with clade 2.3.4.4b viruses have been characterized by unusually high morbidity and mortality [40].

The IAV (H7N9) virus, by contrast, is closely associated with live bird markets (LBMs), which act as amplification hubs for viral transmission. Phylogenetic studies suggest that IAV (H7N9) emerged through re-assortment in wild birds before establishment in domestic poultry. LBMs have been crucial in maintaining AIV circulation and facilitating cross-species viral spillover, particularly given the frequent detection of IAV (H9N2) and IAV (H7N9) viruses in these environments [41].

Transmission of IAV (H5N1) and IAV (H7N9) to humans primarily occurs through direct or indirect contact with infected birds or contaminated environments, especially via exposure of mucous membranes (eyes, nose, mouth) to infected secretions [42].

The expanding host range of clade 2.3.4.4b viruses to include domestic and wild mammals significantly raises concerns about further zoonotic transmission events. Although bird-to-human transmission remains infrequent, the unprecedented host range and ecological reach of recent outbreaks substantially increase the likelihood of viral persistence, evolution, and eventual adaptation to humans [43].

## 3. Evolution of IAV (H5N1) and IAV (H7N9)

The evolutionary dynamics of IAV (H5N1) and IAV (H7N9) viruses are central to understanding their epidemiology, influencing their geographic distribution, pathogenicity, host range, and potential for zoonotic transmission [44]. Both viruses belong to the *Orthomyxoviridae* family and the genus *Alphainfluenzavirus*, characterized by segmented genomes composed of eight single-stranded RNA segments [2].

### 3.1. Genetic Characteristics

#### 3.1.1. Genomic Differences Between IAV (H5N1) and IAV (H7N9)

IAV (H5N1) and IAV (H7N9) viruses are classified by their specific hemagglutinin (HA) and neuraminidase (NA) surface glycoproteins—H5 and N1 for IAV (H5N1), and H7 and N9 for IAV (H7N9) [18,19].

The IAV (H5N1) viruses, particularly those of the *A/goose/Guangdong/1/1996* (Gs/Gd) lineage, have diversified extensively into numerous clades and subclades [7]. In contrast, the emergence of IAV (H7N9) was a distinct event resulting from multiple reassortment processes. Phylogenetic analyses reveal that IAV (H7N9) originated through the acquisition of internal genes from endemic IAV (H9N2) viruses, with its HA and NA genes likely derived independently from wild avian species (Table 3). Notably, the internal gene cassette inherited from IAV (H9N2) significantly contributes to the mammalian pathogenicity of IAV (H7N9) (Figure 1) [7].

After emergence, IAV (H7N9) viruses successfully established domestic chickens as the principal reservoir, facilitating their amplification and dissemination [45].

**Table 3 microorganisms-14-00012-t003:** Genomic Characteristics and Key Mutations of IAV (H5N1) and IAV (H7N9).

FEATURE	IAV (H5N1)	IAV (H7N9)
Primary Strain Studied	A/Vietnam/1194/2004 (human isolate, clade 1; GenBank: AY651333) [46]	Multiple human isolates from 2013 onward (not specified in merged source) [14,35]
HA Receptor Preference (Wild-type)	Binds α2-3–linked sialic acid(avian-type); limited binding to human tracheal epithelium [46]	Binds α2-3–linked sialic acid; some human isolates show dual/shifted tropism [14,47]
Key HA Mutations Conferring Human Airway Binding	G228S: Enables binding toboth ciliated and non-ciliatedhuman tracheal cells (human-like tropism) Q226L + G228S: Strongest human-like binding (mimics H3N2) S227N: Weak dual binding to synthetic α2-6/α2-3; no binding to non-ciliated cellsin tissue Q226L: Loss of α2-3 binding; no gain of human receptor binding [46]	A452T(HA) E64K(HA2) Multibasic cleavage site insertion (in highly pathogenic H7N9) [14,45]
HA Binding Assay Model	Human airway epithelium (HAE) cultures + ex vivo human tracheal tissue sections [46]	Key internal protein adaptations, including the mammalian-adaptive PB2-E627K mutation, contribute to increased virulence [48]
Codon Barrier for Key Mutation	G228 encoded by GGA (93%) or GGG (7%) → G228S requires 2 nucleotide changes → low evolutionary likelihood [46]	Key adaptive HA mutations (e.g., at positions 186 and 226) are often achievable through single-nucleotide polymorphisms, lowering the evolutionary barrier to human adaptation [47,49].
Non-Viable/Dead-End Intermediates	G228R: No binding to synthetic or tissue receptors; virus not rescuable G228A: Retains avian binding; no human tropism [46]	Not reported in H5N1 study
Other Key Proteins and Mutations	(Study focused exclusively on HA) [46]	Polymerase (PB2): E627K, D701N (enhanced mammalian replication) [48].
		Immune Modulation (NS1): Mutations (e.g., V178I) antagonizing interferon response [15].
		Other Internal Proteins: Adaptations in NA, NP, M1 linked to virulence [14]
Host Systems Tested	Human tracheal tissue, HAE cultures, synthetic glycans (3SLN/6SLN) [46]	Mice (primary); chickens (for HA cleavage site); ferrets (PA-X)
Biological Implication	Despite demonstrable potential to adapt to human airways, High genetic barrier (multi-nucleotide changes) limits emergence of transmissible H5N1 [46]	Broad mutational repertoire. Adaptations across multiple gene segments facilitate mammalian infection and severity, contributing to high zoonotic risk [14,15]

Consequently, the wildlife-driven ecology of H5N1 and the live-market-driven ecology of H7N9 necessitate fundamentally different outbreak response strategies.

#### 3.1.2. Mechanisms of Antigenic Drift and Antigenic Shift Between IAV (H5N1) and IAV (H7N9)

Influenza A viruses (IAV) evolve through two principal mechanisms:Antigenic Drift: Antigenic drift involves the gradual accumulation of point mutations—particularly non-synonymous mutations—in the HA and NA genes. These mutations alter critical antigenic sites, allowing the virus to evade host immune responses (Table 3). Antigenic drift is an ongoing process observed across IAV (H5N1) clades and IAV (H7N9) lineages, driving continuous viral evolution and contributing to epidemic persistence [50].Antigenic Shift: Antigenic shift refers to the mixing of gene segments between two or more influenza viruses co-infecting the same host cell. This process, enabled by the segmented nature of the influenza genome, can give rise to novel viral genotypes with new HA and/or NA combinations. The emergence of IAV (H7N9) exemplifies antigenic shift (Table 3), originating from multiple reassortment events involving H7, N9, and IAV (H9N2) viruses. Similarly, reassessment has played a pivotal role in the evolution of Gs/Gd-lineage IAV (H5N1) viruses. IAV (H9N2) viruses, in particular, have been major donors of internal gene segments to several zoonotic influenza strains, including IAV (H5N1) and IAV (H7N9) [7,51].

Recent surveillance has also documented the spread of IAV H9N2 into new regions such as Myanmar and Bangladesh, highlighting its continued role in the genetic diversification and geographic expansion of avian influenza viruses in Asia [52].

### 3.2. Adaptation to Human Hosts

AIVs are naturally adapted to wild aquatic birds, preferentially binding to α2,3-linked sialic acid (SA) receptors in the avian gut, whereas human-adapted influenza viruses target α2,6-linked SA receptors predominantly in the human upper respiratory tract [53]. For avian influenza viruses to achieve efficient human infection and transmission, they typically require specific amino acid substitutions in the hemagglutinin (HA) receptor-binding site—such as G186V (H3 numbering)—that shift or broaden receptor specificity toward human-type receptors [49,54].

As introduced in Section 1.2, adaptation to human hosts requires overcoming fundamental virological barriers. The following section details the specific mutations in IAV (H5N1) and IAV (H7N9) associated with this adaptive process.

Key Mutations Enhancing Mammalian Adaptation:i.IAV (H5N1) Adaptations:Adaptation of IAV (H5N1) to human hosts involves several key genetic changes:HA mutations, such as Q222L and G224S, promote a shift from α2,3- to α2,6-SA receptor binding [10,55].Additional mutations at HA positions 129 and 134 can further modulate binding affinity.Polymerase complex mutations, notably E627K, D701N, and S714R in PB2, enhance viral replication efficiency in mammalian cells [56].Mutations in NA can improve viral fitness by enhancing replication in human airway epithelial cells and facilitating immune escape.
ii.IAV (H7N9) Adaptations:The IAV (H7N9) virus has similarly exhibited key adaptations:Early isolates demonstrated dual receptor specificity, with mutations such as G186V and Q226L in HA facilitating α2,6-SA binding [47].The PB2-E627K substitution has been implicated in increased polymerase activity and replication in mammals.Additional mutations in the NP, M, PA, and NA genes contribute to enhanced virulence, replication efficiency, and modulation of the host immune response [48].NS1 protein mutations have been associated with antagonism of interferon responses, supporting viral persistence in human hosts.

While these adaptations increase zoonotic potential, neither IAV (H5N1) nor IAV (H7N9) has yet acquired the ability for efficient or sustained human-to-human transmission [39].

### 3.3. Antigenic Diversity and Vaccine Challenges

The significant genetic and antigenic evolution observed in IAV (H5N1) and IAV (H7N9) viruses presents formidable challenges for vaccine development and pandemic preparedness.

Impact of Viral Diversity on Vaccine Development

IAV (H5N1): Gs/Gd-lineage IAV (H5N1) viruses have diversified into numerous clades and subclades, each with distinct antigenic properties. The currently dominant clade, 2.3.4.4b, has achieved near-global distribution through wild bird migrations. The continuous emergence of new variants necessitates regular updating of vaccine candidates to ensure antigenic match, complicating mass vaccination efforts [7].IAV (H7N9): Over successive epidemic waves, IAV (H7N9) viruses have evolved significant genetic and antigenic diversity. For example, vaccines developed based on first-wave strains exhibited poor cross-reactivity against fifth-wave viruses, requiring the development of updated candidate vaccine viruses. Regional diversification and reassortment with IAV (H9N2) and other viruses further complicate vaccine design [7].

Additionally, the high rates of antigenic drift, especially within HA and NA proteins, and reassortment with co-circulating avian viruses such as IAV (H9N2), underscore the need for continuous surveillance and the development of broadly protective or “universal” influenza vaccines.

### 3.4. Current Vaccines and Their Limitations

Vaccination remains a cornerstone of pandemic preparedness against zoonotic influenza viruses like IAV (H5N1) and IAV (H7N9). Despite substantial advances, several challenges hinder the full effectiveness of current vaccine strategies [57].

#### 3.4.1. Type Vaccination

Poultry Vaccination: Vaccination in the poultry population is practiced in many countries. A bivalent inactivated H5/H7 vaccine for chickens was introduced in China, and multiple strategies, including this vaccine, seem to have been quite successful against emerging IAV (H7N9) viruses. A vaccine against one clade of virus may not protect against other clades or subclades due to antigenic differences. Vaccinating cattle to reduce transmission is also being explored, with research teams in the early stages of developing vaccines for livestock [38].Human Vaccination: Development and deployment of effective vaccines against HPAI (H5N1) are of paramount importance. Several IAV (H5N1) vaccines have received approval, including Audenz (FDA, 2020) for individuals aged six months and older at increased risk, Sanofi Pasteur’s vaccine (FDA, 2007) for individuals aged 18–64 at elevated risk [58], Some countries, like Finland and Austria, have made H5 vaccines available to individuals with higher exposure risks like farm workers as a precaution (Table 4). Strategic deployment of vaccination is needed to minimize human cases. However, existing stockpiles may be insufficient in a pandemic scenario [59].

Recent advances include the deployment of bivalent H5/H7 poultry vaccines in China and the development of recombinant and mRNA-based human vaccine candidates. Given the establishment of IAV (H5N1) in dairy cattle, vaccination trials for livestock are under early evaluation. A summary of currently available and experimental vaccines is presented in Table 4.

#### 3.4.2. IAV (H5N1) Vaccine Candidates and Limitations

i.Approved Vaccines

Several IAV (H5N1) vaccines have been licensed, primarily for pandemic stockpiling or high-risk occupational groups [68]:


Audenz (FDA, 2020): An adjuvanted, cell-culture-derived IAV (H5N1) vaccine for individuals aged six months and older.Sanofi Pasteur IAV (H5N1) Vaccine (FDA, 2007): Approved for adults aged 18–64 at elevated risk.GSK Adjupanrix and Prepandemic Vaccines (EU-approved): Based on A/VietNam/1194/2004 (IAV (H5N1)) strain.Seqirus Vaccines (Celldemic, Incellipan): EMA-approved for avian influenza preparedness.CSL Seqirus H5N8 Vaccine: Targets clade 2.3.4.4b HA, with N8 NA, using an MF59 adjuvant.


Research is also advancing alternative platforms, including recombinant HA vaccines, live-attenuated formulations, and mRNA vaccines.

LIMITATIONS


Antigenic Drift: Rapid viral evolution necessitates frequent updates to vaccine strains.Cross-Protection Gaps: Vaccines targeting one clade may not be effective against others.Low Immunogenicity: Most IAV (H5N1) vaccines require adjuvants and prime-boost regimens.Limited Field Data: Real-world effectiveness in preventing disease or transmission remains unproven.Stockpile Constraints: Current global vaccine reserves are insufficient for large-scale deployment.


Low Uptake: Vaccination rates among high-risk occupational groups remain poor.Animal Model Limitations: Predictive value of mRNA vaccine studies in ferrets has not translated reliably to humans.

#### 3.4.3. IAV (H7N9) Vaccine Candidates and Limitations

Poultry and Human Vaccination Efforts [69].

Poultry: China introduced a bivalent inactivated H5/H7 vaccine, reducing human cases post-wave 5, though sterilizing immunity remains elusive.Human Vaccines: Multiple candidates have been developed:
i.Inactivated Vaccines: Require high doses and adjuvants for robust protection.ii.Live-Attenuated Vaccines (LAIVs): Offer stronger T-cell responses and heterosubtypic protection potential.iii.Virus-Like Particles (VLPs): Show promise for safety and immunogenicity in early trials.

LIMITATIONS

Antigenic Drift and Evolution: Ongoing viral changes require frequent updates to vaccine compositions.Diverse Viral Lineages: Genetic variability complicates the design of broadly protective vaccines.Need for Universal Vaccines: Broader cross-protection strategies remain a critical unmet need.Suboptimal Immune Responses: Current vaccines often fail to stimulate sufficient neutralizing antibody and NA-specific responses [57].Persistent Threat: Despite decreased human cases, IAV (H7N9) and related viruses continue to circulate in poultry and wild birds, necessitating ongoing vigilance.

## 4. Clinical Manifestations and Diagnosis

### 4.1. Symptoms and Disease Progression

Equally HPAI (H5N1) and (H7N9) viruses can cause significant illness in humans, often presenting initially with symptoms characteristic of typical influenza. However, they can rapidly progress to severe disease [70].

Common Symptoms:

i.Initial symptoms often include influenza-like illness such as fever (including high fever), cough, muscle or body aches (myalgia), headaches, and fatigue [31].ii.Other symptoms can include sore throat and shortness of breath or difficulty breathing (dyspnea).iii.Diarrhea has been reported in both infections, occurring in more than 50% of patients in one IAV (H5N1) series from Vietnam but less than 10% in others. It is listed as a less common symptom for IAV (H5N1) and among typical flu symptoms for IAV (H7N9).iv.Nausea and vomiting can also occur, sometimes following initial flu-like symptoms in IAV (H5N1) [31].

Ocular manifestations such as conjunctivitis have been linked to direct exposure of conjunctival epithelium expressing α2,3 sialic-acid receptors, permitting localized viral replication. Conversely, severe pulmonary disease results from viral tropism for α2,3 receptors in the lower respiratory tract, causing diffuse alveolar damage, high viral loads, and cytokine-mediated inflammation [10].

### 4.2. Differences in Clinical Presentation and Complications

Both viruses can cause a wide spectrum of illness in humans, ranging from mild infections to severe, life-threatening disease involving multiple organs and systems.

i.Severity: Historically, IAV (H5N1) was often associated with severe disease and had a high case fatality rate (CFR), estimated at 48% among reported cases since 2003. IAV (H7N9) also has a notable CFR, though it varies depending on the time period and location, with one report noting a nationwide CFR of 39.6% compared to ~13.3% in Shenzhen, China [9], potentially due to intervention measures. Recent human infections with IAV (H5N1) clade 2.3.4.4b have frequently been clinically mild and self-resolving, although severe and fatal cases still occur globally. The reasons for the variation in outcome are likely multi-factorial, including virus genotype, duration and route of exposure, viral load, individual health status, personal protective measures, and medical treatment [71].ii.Extrapulmonary Manifestations: IAV (H5N1) infection more often leads to extrapulmonary manifestations compared to pandemic influenza viruses. These can include liver impairment with elevated transaminases, particularly in severe cases. Notably, IAV (H5N1) infections are characterized by a heightened inclination towards Central Nervous System (CNS) disease manifestation when contrasted with seasonal influenza. In animal models, IAV (H5N1) can access the CNS via olfactory and trigeminal nerves, causing severe meningoencephalitis (Table 5). In animals generally, HPAIV clade 2.3.4.4b spontaneous infections are characterized by remarkable neurotropism and systemic virus spread [71].iii.Immune Response: Severe IAV (H7N9) and IAV (H5N1) infections are associated with high inflammatory cytokine and chemokine levels in the lungs and peripheral blood, referred to as hypercytokinemia or cytokine storm. While potentially correlated with severe disease, hypercytokinemia levels caused by IAV (H7N9) may be lower than those caused by H5N6. HP-IAV (H7N9) seemed to induce higher cytokine levels than LP-IAV (H7N9), but the difference was not significant in one study [72,73].iv.Fatal Outcomes: In both infections, fatal outcomes are often associated with the development of complications such as acute respiratory distress syndrome (Table 5), severe pneumonia, and multi-organ failure (including respiratory and renal failure, pulmonary hemorrhage, pneumothorax, and pancytopenia). For IAV (H5N1), fatal outcomes have been associated with high viral loads, lymphopenia, and elevated inflammatory cytokines and chemokines. For IAV (H7N9), multiple organ failure is a major cause of death. Nosocomial bacterial infections, often with antibiotic-resistant organisms, are also common in severe IAV (H7N9) cases [15].

### 4.3. Diagnostic Tools

Timely and accurate diagnosis is critical for minimizing morbidity and mortality and reducing pandemic potential. Various samples are used, including respiratory, rectal specimens, and serum. For IAV (H5N1) [57], throat swabs and lower respiratory specimens are preferred over nasopharyngeal swabs due to the virus’s predilection for the lower tract. For recent US IAV (H5N1) cases, conjunctival swabs showed high positivity rates, especially in patients with conjunctivitis [74].

#### RT-PCR and Real-Time PCR

Timely and accurate diagnosis of IAV (H5N1) and IAV (H7N9) infections in humans and animals relies on the availability of both molecular and serological methods. Molecular assays (e.g., RT-PCR and real-time RT-PCR) are essential for early clinical diagnosis [75,76], enabling rapid case confirmation and infection control measures. In contrast, molecular tools used for epidemiological purposes—such as whole-genome sequencing, HA/NA subtyping [77,78], or clade classification—serve surveillance and risk-assessment objectives and typically require more advanced laboratory infrastructure. These two applications differ not only in their finality (clinical vs. public health) but also in the technical specifications and equipment needed. Serological methods (e.g., HI, ELISA, MN assays) remain valuable for retrospective confirmation [73,79], seroprevalence studies, and vaccine response evaluation, but are generally unsuitable for acute diagnosis due to delayed antibody kinetics.

### 4.4. Treatment Options

Prompt initiation of antiviral therapy upon illness onset plays a critical role in reducing the mortality rate.

#### 4.4.1. Neuraminidase Inhibitors (NAIs)

Oseltamivir (Tamiflu^®^) and zanamivir (Relenza^®^) are currently the primary treatments for IAV (H5N1) virus infections. Peramivir is also mentioned as an NAI. NAIs work by preventing the virus from exiting cells and spreading [71]. Prompt initiation of therapy is associated with reduced symptom duration, decreased reliance on antibiotics, and quicker recovery in seasonal influenza. For IAV (H5N1), observational studies and clinical guidelines support the recommendation for oseltamivir treatment. Most recent U.S. IAV (H5N1) cases received oseltamivir, often within 48 h of symptom onset [34].

#### 4.4.2. Polymerase Inhibitors

Baloxavir marboxil, a cap-dependent endonuclease inhibitor targeting the polymerase acidic (PA) protein, is a newer antiviral agent with demonstrated efficacy against influenza A and B viruses, including avian subtypes H5N1 and H7N9 [80].

#### 4.4.3. Adamantanes

These agents impede the initial phases of viral replication. They were used in the initial IAV (H5N1) outbreak in Hong Kong in 1997, but resistance is now widespread due to M2 protein mutations. Consequently, adamantanes are presently not recommended for treating *influenza A virus* due to elevated resistance levels, although susceptible strains are limited to some older clades [81].

#### 4.4.4. Resistance

While most IAV (H5N1)) viruses from recent US human infections are susceptible to currently available antiviral agents, some viruses have shown mutations conferring minor decreases in susceptibility to neuraminidase inhibitors and baloxavir. The effectiveness of NAIs can be compromised by viral evolution and mutations, such as H274Y in the NA gene, which confers resistance to oseltamivir [82].

#### 4.4.5. Supportive Care

In severe cases, management includes comprehensive measures such as mechanical ventilation, corticosteroids, antibiotics (to treat secondary bacterial infections), and fluid infusion.

#### 4.4.6. Future Directions

Further exploration of antiviral therapeutics and treatment strategies is essential. Research should focus on identifying and developing novel antiviral agents with broad-spectrum activity against AIVs. Investigating host immune responses and developing immunomodulatory therapies could also aid in reducing disease severity and improving patient outcomes [83,84].

## 5. Global Response and Public Health Measures

Addressing the threats posed by HPAI (H5N1) and IAV (H7N9) viruses requires a multi-faceted approach encompassing robust surveillance, proactive prevention strategies, and strong international collaboration. These measures are critical for minimizing morbidity and mortality, reducing the potential for pandemics, and managing the impacts on wildlife, agriculture, and human health [85].

### 5.1. Surveillance Systems

Strengthening global surveillance systems for avian influenza is crucial for promptly identifying and monitoring emerging strains, particularly those with zoonotic potential. This involves continuous monitoring of wild bird populations, domestic poultry, and high-risk areas. Timely and accurate diagnosis is a critical component of surveillance for minimizing morbidity and mortality in humans [86].

Various Types of Surveillance Are Employed:

i.Monitoring of animal populations: This includes surveillance data and mortality data from poultry and wild birds. For North America, this data is collated from sources like USDA-APHIS, USGS-WHISPers, and CFIA/ACIA. Large-scale animal disease data is managed by various authorities including local, state, indigenous, federal, and transnational agencies, as well as non-governmental research groups. The FAO’s EMPRES-i+ database is a primary collation of international regional/country-level data, capturing approximately 30% of IAV (H5N1) disease events. WOAH-WAHIS is another reporting system used globally [1,87].ii.Human case surveillance: Countries should remain vigilant for potential human cases of avian influenza, especially in geographic areas where the virus is highly circulating in poultry, wild birds, or other animals. Healthcare workers in these areas need to be aware of the epidemiological situation and the range of symptoms associated with avian influenza infection in humans. Surveillance systems for seasonal influenza have also highlighted the importance of typing viruses to detect zoonotic avian influenza cases. In the U.S., state and local public health officials have monitored occupationally exposed persons for symptoms after exposure to potentially infected animals and collected specimens from symptomatic persons. Most recent U.S. cases were identified through symptom monitoring. Monitoring of bovine veterinary practitioners for HPAI A(H5) infections has also occurred [88].iii.Wastewater surveillance: This is another method being used, with the CDC detecting the presence of *influenza A virus* in wastewater in several U.S. states and cities. Virome sequencing has also identified HPAI (H5N1) clade 2.3.4.4b in wastewater. However, identifying specific subtypes at the national level may be challenging with current CDC methods, and the source (avian, bovine, or human) can be uncertain [53].iv.Genetic surveillance: Ongoing genetic analysis and monitoring are needed due to genetic variations among HPAI (H5N1) strains. Studying mutations and reassortment events provides insights into the virus’s evolutionary potential, transmission dynamics, and pathogenicity [89].v.Serological surveillance: Detection of IAV (H5N1)-specific antibodies is essential for epidemiological investigations. Serological studies, combined with mortality data, can help infer levels of flock immunity in different species. Testing for seroprevalence in healthy populations can be performed using MN or HI tests, though HI has limited value for detecting antibodies against avian viruses in humans/mammals due to low sensitivity [90]. Serology testing was conducted in one U.S. state for IAV (H5N1).

Challenges exist in understanding changing disease dynamics due to limitations of detection, testing, reporting, and collation across national and international organizations. Not all sick and dead wild birds detected and reported are captured in databases. Surveillance efforts and laboratory characterization are fundamental to assess the threat posed by emerging H5 and H7 viruses [20].

### 5.2. Prevention Strategies for HPAI Outbreaks and Transmission

A range of strategies are employed to prevent HPAI outbreaks and transmission to humans and other animals [91]:i.Minimizing Exposure: The most effective approach to prevent human IAV (H5N1) infection is to minimize exposure to potential sources of the virus. This includes avoiding contact with dead, sick, or abnormal birds and mammals unless properly trained and equipped. Caution with pets is also important [92].ii.Protection for Exposed Individuals: Public health efforts focus on protecting workers exposed to potentially infected animals. This includes the implementation of prevention measures on farms, including PPE use. Despite the importance, low rates of PPE use have been noted among dairy workers [91].iii.Biosecurity Measures: Improving biosecurity measures is a priority for the poultry industry to reduce transmission to and within domestic poultry. Good farm biosecurity is paramount. This includes implementing policies for visitors [92].iv.Live Poultry Market (LPM) Interventions: LPMs have been identified as major sources and hotspots for avian influenza outbreaks and human infection. Closure of live poultry markets has been highly effective in reducing the risk of IAV (H7N9) infection in humans. Other effective interventions in the LPM system include rest days and banning live poultry overnight. Enhanced disinfection and regular closure of wet markets also reduced transmission risk [92].v.Poultry culling, movement restrictions, biosecurity, and live bird market interventions—which constitute the primary and most effective preventive strategies—are now presented as the central focus [3].vi.Vaccination content has been significantly condensed in the prevention section and relocated to a dedicated subsection (e.g., “Vaccination Strategies”) where its role, limitations, and current recommendations (including the absence of routine human vaccine use) are discussed in appropriate context [38,59].vii.Adaptive Management: Effective disease management responses depend on improved systems for decision-making, particularly in the face of uncertainty. Formal methods of decision analysis can aid in allocating scarce resources and prioritization of scientific inquiry to inform management and conservation actions. Management-driven scientific inquiry is urgently needed to establish recommended disease response protocols [93].
Addressing Underlying Factors: In the longer term, addressing underlying factors such as intensive farming practices and wildlife trade, which create environments conducive to viral mutations, is crucial in preventing pandemics [3].
Pasteurization: The U.S. Department of Agriculture (USDA) and the Centers for Disease Control and Prevention (CDC) emphasize the critical need for pasteurization of dairy products to mitigate the risk of human infection [94].



### 5.3. International Collaboration

Addressing the challenges posed by HPAI (H5N1) necessitates a proactive approach involving strengthening international collaboration and information sharing among countries [95,96].

#### 5.3.1. One Health Approach

A cohesive One Health approach encompassing wildlife, poultry, and humans is crucial for addressing disease dynamics, management response, and reducing future negative impacts. This multisectoral approach recognizes the interconnectedness of the health of people, animals, plants, and the environment. Efforts must involve collaboration across human, animal, and environmental health sectors [95,96].

#### 5.3.2. Information Sharing and Coordination

Establishing robust communication networks, sharing surveillance data, and coordinating response efforts will facilitate rapid response, early containment, and effective control of outbreaks. The timely sharing of surveillance data and whole genome sequences of isolated viruses has enabled scientists globally to develop virus detection protocols, diagnostic tests, and vaccines. A global network for real-time data sharing and coordination is needed. International cooperation has been proven successful in controlling outbreaks, such as the initial IAV (H7N9) outbreak in China [95].

#### 5.3.3. Involvement of International Organizations

Key international organizations are involved in coordinating global response efforts. These include [18,94,95]:i.WHO (World Health Organization).ii.FAO (Food and Agriculture Organization of the United Nations).iii.WOAH (World Organisation for Animal Health).iv.ECDC (European Centre for Disease Prevention and Control).v.EFSA (European Food Safety Authority).

These organizations contribute through various activities, including joint assessments of public health risk, providing guidance for investigation and management of outbreaks, producing scientific reports and overviews of the situation, and advocating for preparedness measures. The WHO, FAO, and WOAH have conducted updated joint assessments of recent influenza IAV (H5N1) virus events [28].

#### 5.3.4. Collaboration Between Agencies

Coordination is needed across multiple scales and between agencies with separate authorities. The Task Force of Joint Prevention and Control System in China, involving different institutes, responded well to the IAV (H7N9) outbreak.

Strengthening surveillance systems, increasing testing capacity, investing in research for vaccines and treatments, and fostering international cooperation are critical steps for pandemic preparedness [57,97].

## 6. Lessons Learned and Future Directions

Addressing the complex and evolving challenge of HPAI requires continuous learning from past outbreaks and proactive planning for future threats [97].

### 6.1. Insights from Past Outbreaks

Past outbreaks of HPAI (H5N1) and (H7N9) have provided critical insights into the dynamics of these viruses and effective response strategies. The unprecedented epizootic of Eurasian origin IAV (H5N1) in North America since late 2021, affecting poultry, wild birds, and mammals across a wide geographic range, highlights a significant shift in viral dynamics, including widespread symptomatic infections in wild birds and changes in seasonality. This event underscores the potential for HPAI to establish endemism in new regions, as has occurred in Asia, Africa, and Europe. The magnitude and complexity of this spread emphasize the need for effective decision framing and improved systems for disease management, particularly at the interface of wildlife, poultry, and human health, recognizing the challenges of complex governance structures and limited resources [97].

The experience with HPAI (H7N9) in China, which caused five epidemic waves of human infections between 2013 and 2017, demonstrated the effectiveness of coordinated public health intervention strategies. A key lesson learned was the high efficacy of closure of live poultry markets (LPMs) in reducing the risk of IAV (H7N9) infection in humans. Other effective interventions within the LPM system included rest days and banning live poultry overnight [98]. The introduction of a bivalent inactivated H5/H7 vaccine for chickens in China also appears to have been quite successful in controlling the IAV (H7N9) epidemic, with only sporadic human cases reported since wave 5. Although the IAV (H7N9) epidemic appears to have gradually disappeared after these waves, continuous surveillance in LPMs and wild birds still detects the virus, indicating a continuous threat. The response to IAV (H7N9) serves as a valuable model for preparedness against other pandemic-related infectious diseases. Both H5 and H7 viruses have characteristics that warrant significant pandemic preparedness efforts, despite factors like HA instability that may limit efficient human-to-human transmission. The Asian lineage of IAV (H7N9) is recognized by the CDC as having the highest potential pandemic risk [21,32].

### 6.2. Emerging Threats by 2025–2026

As of 2025, the threat posed by HPAI (H5N1) remains significant and dynamic. There is high uncertainty about the future trajectory of HPAIV disease dynamics, but the widespread geographic range and the diversity of affected species in the Americas strongly suggest the likelihood of future endemism. Based on past patterns, a continued increase in transmission and mortality in North American birds is anticipated [21,99]. The current situation with clade 2.3.4.4b HPAI (H5N1) is concerning and presents a significant risk of an influenza pandemic [10]. The overall number of HPAI virus detections in birds in the current 2024–2025 epidemiological year has already surpassed that of the previous year.

A particularly troubling emerging threat is the establishment of HPAI (H5N1)) virus in dairy cattle in the United States. This marks a new chapter, with alarming expansion across multiple states and affected herds as of September 2024. The potential for a pandemic presence in cattle could turn dairy farms into viral reservoirs, facilitating further evolution and increasing the risk of human transmission. Mutations associated with enhanced replication in mammals, such as the PB2 E627K mutation, have already been identified [36]. Detection of the virus in the central nervous system of infected animals, including cats, highlights the virus’s broad tissue tropism and severe pathogenic potential. Beyond cattle, poultry outbreaks continue sporadically, and the persistent circulation of HPAI in wild birds in Europe is expected to continue for many years, presenting an ongoing risk to poultry. H5 viruses also remain endemic in poultry in the Middle East, and IAV (H7N9) continues to circulate in poultry in China, maintaining a threat [59].

Recent epidemiological data confirm that HPAI (H5N1) has reached a panzootic state, affecting avian and mammalian hosts across nearly all continents. This widespread and sustained circulation underscores the shift from regional epizootics to a globally entrenched disease dynamic, with significant implications for animal and public health surveillance systems [18,100].

### 6.3. Recommendations for 2025–2026 and Beyond

Responding effectively to the ongoing and future threats of HPAI requires a comprehensive, proactive, and coordinated approach. A crucial recommendation is the need for improved decision-making, utilizing formal analytical methods like Structured Decision-Making (SDM). This framework can help management agencies and scientists frame the complex problem, articulate objectives, generate alternatives, predict consequences, and make trade-offs, particularly in the face of high uncertainty and limited resources [97]. Decision analysis can also help priorities scientific inquiry to address critical knowledge gaps. Coordination across multiple scales, from local properties to global migratory connections, and between agencies with separate authorities is essential for effective response [28,95].

Specific Recommended Actions Include:

i.Strengthening Surveillance: Strengthening global surveillance systems for avian influenza is crucial for promptly identifying and monitoring emerging strains, especially those with zoonotic potential. This involves continuous monitoring of wild bird populations, domestic poultry, and high-risk areas. Surveillance should also aim to detect changes in virus pathogenicity and transmissibility and identify potential mammal-to-avian transmission [20,88].ii.Enhancing Genetic Monitoring: Ongoing genetic analysis and monitoring of viral evolution is needed due to genetic variations, studying mutations and reassortment events to understand evolutionary potential, transmission dynamics, and pathogenicity. Genetic characterization should be reinforced, especially in areas with mammalian infections. Monitoring viral evolution for changes that stabilize the HA molecule is particularly important [50,55,89].iii.Advancing Vaccination Strategies: Development and deployment of effective vaccines against HPAI (H5N1) are of paramount importance. Research should focus on improving vaccine efficacy, broadening protection across strains, and optimizing delivery. Vaccination should be considered in regions where poultry are not vaccinated to protect production and reduce human exposure risk. Controlling the dairy cattle outbreak may require vaccination of cows or better infection control. Vaccination for animals in fur farms and enhanced biosecurity measures are also suggested [27].iv.Improving Therapeutic Options: Invest in the development of novel antiviral agents and immunomodulatory therapies. And Monitor for antiviral resistance and adapt treatment guidelines accordingly [83,84].v.Strengthening Biosecurity: Implementing stringent biosecurity measures and prevention strategies to minimize exposure to potential sources of the virus. This includes avoiding contact with dead, sick, or abnormal birds and mammals unless properly trained and equipped, using appropriate Personal Protective Equipment (PPE) for workers exposed to potentially infected animals, implementing adequate biosafety and biosecurity at occupational settings, and improving biosecurity measures on farms [91,92].vi.Building International Collaboration: Foster robust information sharing, surveillance coordination, and joint response mechanisms across countries and sectors. And encourage real-time sharing of viral sequences and outbreak data [85,95].vii.Applying Structured Decision-Making (SDM): Utilize SDM frameworks to guide resource allocation, prioritize research, and develop effective outbreak management protocols under uncertainty [97].viii.Adopting a One Health Approach: Adopting a cohesive One Health approach encompassing human, animal, and environmental health sectors is crucial for addressing disease dynamics and managing responses effectively. EFSA, ECDC and WHO have provided practical guidance for managing zoonotic outbreaks using this interdisciplinary approach [95].ix.Addressing Root Causes: Tackle systemic risk factors, including intensive animal farming, wildlife trade, and ecosystem disruption, which drive viral emergence and cross-species transmission [95].

The current outbreak serves as a stark reminder that effective and proactive response is necessary to avert potential global health crises. Surveillance efforts and laboratory characterization remain fundamental to assessing the threat posed by emerging H5 and H7 viruses [97].

## 7. Conclusions

Highly pathogenic avian influenza (HPAI), particularly IAV (H5N1) and IAV (H7N9) viruses, continues to pose a dynamic and significant threat to global health. Insights from past outbreaks and assessment of current and emerging risks reinforce the urgent need for sustained vigilance, coordinated action, and strategic preparedness [91,92].

HPAI H5N1 has entered a panzootic phase, with ongoing transmission among wild birds, poultry, and multiple mammalian species worldwide. Together with the residual threat of IAV (H7N9), these viruses continue to pose a dynamic and evolving risk to global health [18].

Lessons learned from previous outbreaks emphasize critical strategies and challenges. The unprecedented spread of Eurasian-origin IAV (H5N1) in North America since 2021—affecting poultry, wild birds, and mammals across broad geographic regions—demonstrates the virus’s ability to establish endemism beyond its traditional range. This event highlighted the complexity of managing disease at the wildlife–poultry–human interface, exacerbated by fragmented governance and limited resources. Similarly, the IAV (H7N9) epidemics in China underscored the effectiveness of targeted interventions, notably live poultry market closures and poultry vaccination, in reducing human infections. Despite these successes, the persistence of both H5 and H7 viruses in animal reservoirs continues to demand robust pandemic preparedness efforts [101].

Looking toward 2026 and beyond, the threats posed by HPAI remain substantial. The widespread geographic distribution, the expanding host range, and the increasing number of detections suggest that clade 2.3.4.4b IAV (H5N1) is likely becoming endemic in the Americas [10,11,18]. The establishment of IAV (H5N1) infections in U.S. dairy cattle—accompanied by mammalian-adaptive mutations such as PB2 E627K and evidence of neuroinvasion—marks a concerning shift that could create new viral reservoirs and facilitate further evolution. Concurrently, sporadic poultry outbreaks, persistent wild bird infections in Europe, and continued IAV (H7N9) circulation in China underscore the sustained global threat landscape.

Effectively addressing these challenges requires a comprehensive, proactive strategy centered on the following:i.Strengthening surveillance systems across wildlife, poultry, cattle, and high-risk human populations to detect emerging strains and critical evolutionary changes.ii.Enhancing genetic monitoring to track mutations linked to pathogenicity, host adaptation, and transmission potential.iii.Accelerating vaccine development and deployment, with a focus on the broader protection, rapid scalability, and inclusion of under-vaccinated regions and sectors.iv.Advancing antiviral research and improving therapeutic strategies.v.Implementing stringent biosecurity and exposure prevention measures at farms, markets, and occupational settings.vi.Fostering international collaboration and data sharing to enable rapid response and containment.vii.Adopting a One Health approach that integrates human, animal, and environmental health responses [96].viii.Investing in research and structured decision-making frameworks to prioritize critical knowledge gaps and guide evidence-based policy actions.ix.Promoting long-term structural changes in farming practices and wildlife trade management to reduce zoonotic spillover risks.x.Ensuring comprehensive testing and pasteurization of dairy products to minimize risks of foodborne transmission.

The ongoing situation serves as a powerful reminder that early, coordinated, and science-driven interventions are essential to avert future global health crises. Continued surveillance, laboratory characterization, and strategic planning will remain foundational to mitigating the risks posed by emerging H5 and H7 viruses.

## Figures and Tables

**Figure 1 microorganisms-14-00012-f001:**
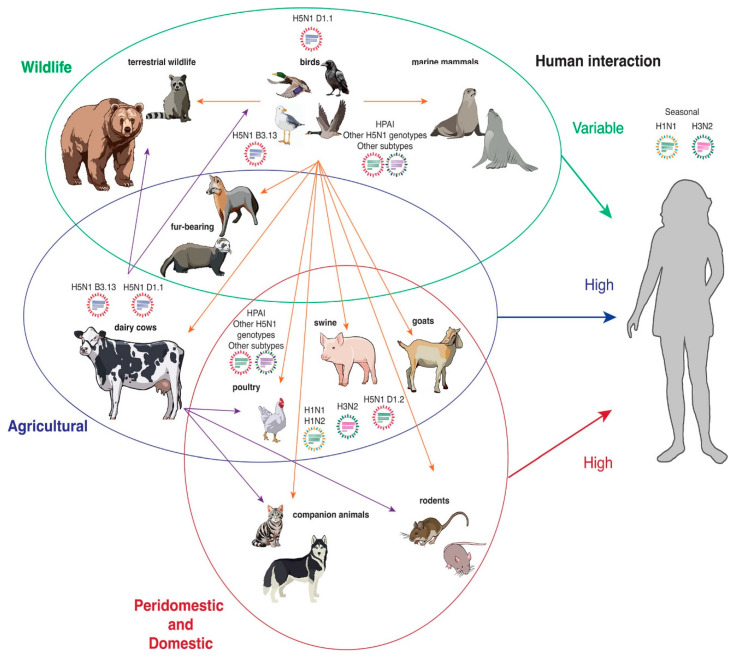
Animal species infected with IAV (H5N1). Wildlife species are encircled in green, agricultural species are encircled in blue, and peridomestic and domestic species are circled in red. Orange arrows depict transmission from an avian host. Purple arrows depict transmission from a bovine host. Icons represent influenza A virus detection in the respective host species. A strains known to circulate in species that present a high risk for re-assortment [9].

**Figure 2 microorganisms-14-00012-f002:**
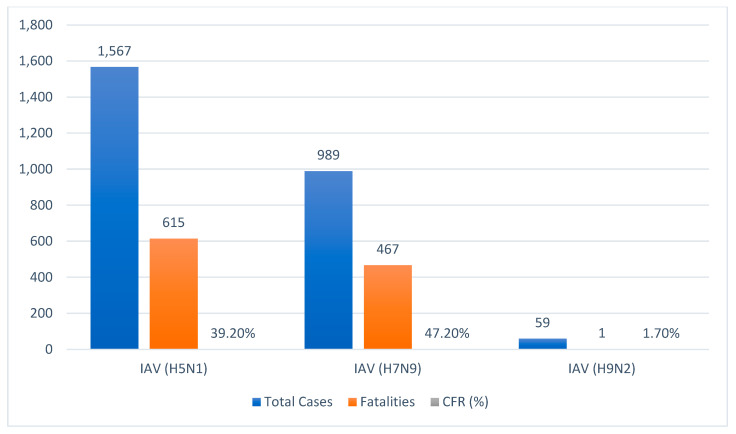
Proportion of Cumulative Human Cases and deaths for IAV (H5N1), IAV (H7N9), and IAV (H9N2).

**Table 4 microorganisms-14-00012-t004:** Vaccine Summary: IAV (H5N1) and IAV (H7N9), Including Animal Vaccines.

Target Species and Subtype	Vaccine Name/Candidate	Type and Composition/Platform	Status and Relevance
Human IAV (H5N1) (Pandemic Preparedness)	AUDENZ [58]	Adjuvanted *Influenza A* (IAV (H5N1)) Monovalent Vaccine (injectable emulsion). Prepared from virus propagated in MDCK cells.	US FDA approved for active immunization in individuals aged six months and older at increased risk of exposure to IAV (H5N1).
Human IAV (H5N1) (Pandemic Preparedness)	CSL Seqirus Vaccine [58]	Classical split virus vaccine that contains the MF59 adjuvant.	Licensed H5 Vaccine with HA matched to Clade 2.3.4.4b (A/Astrakhan/3212/2020), although the NA component is the N8 subtype (mismatched). Licensed based on immunogenicity data.
Human IAV (H5N1) (Pandemic Preparedness)	Celldemic, Incellipan [58]	Vaccines designed for active immunization against avian influenza (specific composition not detailed in sources).	EMA Recommended/Approved (vaccines part of the EU’s pandemic preparedness strategy) authorized in April 2024.
Human IAV (H5N1) (R&D)	mRNA Vaccines [58]	Nucleoside-modified mRNA vaccine.	Development ongoing. mRNA-based IAV (H5N1) vaccines targeting Clade 2.3.4.4b have shown to induce significant protection in ferrets. HHS invested $176 million in Moderna for development.
Human IAV (H7N9) (Cvv-Lpaiv)	A/Anhui/1/2013-like strain, A/Hunan/2650/2016 strain [60]	Candidate Vaccine Virus (CVV) based on Low Pathogenic Avian Influenza Virus (LPAIV).	Recommended by WHO for pandemic preparedness; the A/Hunan/2650/2016 strain was suggested to replace the A/Anhui/1/2013 CVV due to poor reaction to antiserum of the older strain.
Human IAV (H7N9) (Cvv-Hpaiv)	A/Guangdong/17SF003/2016-like strain [60]	Candidate Vaccine Virus (CVV) based on Highly Pathogenic Avian Influenza Virus (HPAIV).	Proposed by WHO as a CVV for HPAIVs.
Human IAV (H7N9) (Vaccine Types)	Various IIVs (Split, Subunit, Whole Virus) [16,37]	Inactivated Influenza Vaccines (IIVs). Most are reassortant viruses with HA/NA from CVV and six internal genes from A/PR/8/34 (PR8 backbone).	Multiple manufacturers (Novartis, Sanofi Pasteur, GSK, Medigen Vacc Corp) have candidates that advanced to human clinical trials. Split vaccines often require oil-in-water adjuvants.
Human IAV (H7N9) (Vaccine Types)	LAIV (Len17-based) [61]	Live Attenuated Influenza Vaccine (LAIV). Generated using classical reassortment techniques.	Tested in clinical trials. LAIVs stimulate cytotoxic CD8+ T lymphocytes, conferring heterosubtypic protection.
Human IAV (H7N9) (Vaccine Types)	VLPs [62]	Virus-Like Particles (VLPs) incorporating HA, NA, and M1 structural proteins. Produced, e.g., in insect cells or plants.	Highly immunogenic, safe, and dose-sparing, often using saponin-based adjuvants (e.g., ISCOMATRIX^®®^). Advanced to human clinical trials.
Poultry (IAV (H9N2))	Traditional Inactivated Vaccines [63]	Traditional inactivated vaccines.	Used in many countries (China, Israel, Egypt, etc.) to combat endemic IAV (H9N2). Sub-optimal use may drive antigenic drift.
Captive Birds (H5)	H5N2 MSD, GALLIMUNE Flu H5N9 [64]	Inactivated HPAI vaccines (implied type, often adjuvanted inactivated vaccines).	Encouraged by EAZA/EAZWV for zoos to vaccinate birds against HPAI A(H5) with the best commercially available product. As of February 2022, these specific vaccines were available in Denmark, France, and Hungary. Anseriformes generally respond best.
Wild Birds (IAV (H5N1))	Unspecified HPAI Vaccine [65]	(Unspecified type)	Emergency use was authorized for California Condors (*Gymnogyps californianus*) in response to IAV (H5N1)-related mortalities.
Wild Mammals	Unspecified HPAI Vaccine [66]	(Unspecified type)	African Penguins (*Spheniscus demersus*) were reported as having been vaccinated against high-pathogenicity avian influenza.
Domestic Mammals (Cattle IAV (H5N1))	Conventional and mRNA Vaccines [67]	Inactivated H5 avian influenza vaccine (tested in calves/cows). Conventional and mRNA vaccines (under development).	Vaccination of dairy cows is currently being discussed in light of the US outbreak, due to the vast economic impact. Experiments are running in the USA to develop an AI vaccine for cattle.

**Table 5 microorganisms-14-00012-t005:** Clinical Features and Complications of IAV (H5N1) and IAV (H7N9) Infections.

Feature	IAV (H5N1)	IAV (H7N9)	Notes
Severity	Severe; high mortality [31,32]	Severe; emerged high pathogenicity in wave 5 [14]	Both associated with ARDS, pneumonia
Common Symptoms	Respiratory and gastrointestinal symptoms, elevated aminotransferase [5]	Respiratory symptoms; some GI involvement [14,15]	Symptoms overlap but vary
Age Distribution	Median 18 years (USA 2024: adults) [33]	Median 63 years [35]	IAV (H9N2) mainly affects young children
Sex Ratio (M:F)	1:1.19 [31]	1:0.45 (male predominance) [35]	Differences in susceptibility noted
Coexisting Conditions	Rare [31,33]	Common (elderly, hypertension, diabetes) [15]	Underlying conditions worsen outcomes in IAV (H7N9)
Transmission	Primarily animal-to-human [5]	Primarily animal-to-human (live bird markets) [35]	Human-to-human transmission rare
Antiviral Sensitivity	Sensitive to NA inhibitors [5,34]	Sensitive to NA inhibitors [15,68]	Treatment feasible with neuraminidase inhibitors
Complications	CNS involvement [71], Multi-organ failure [5]	Nosocomial infections [15], elevated Angiotensin II	Severe systemic complications in both

## Data Availability

No new data were created or analyzed in this study. Data sharing is not applicable to this article.

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
