# Peer review of "Highly Pathogenic Avian Influenza: Tracking the Progression from IAV (H5N1) to IAV (H7N9) and Preparing for Emerging Challenges"

_microorganisms, 2025, doi:10.3390/microorganisms14010012_

Round 1
Reviewer 1 Report (Previous Reviewer 1)
Comments and Suggestions for Authors
This paper remains unacceptable. It is not seriously written and still errors and omissions have been noted.
The abbreviation AIV (Avian influenza virus), for example, may be used through out the text each the viruses are cited: it should be used before eachgiven serotype: e.g. IAV(H5N1).
Virus name, species, genus have to be written in italics.
The numbering of the reference is utterly unacceptable as it does not correspond to their order of apparition in the text as recommended. In tables 1 to 5: References should be inserted in the table (to indicate them in brackets in the title is not acceptable as their accuracy is not verifiable).
Moreover references do not always correspond to what is written; e.g.: Line 133: reference [14] is about bat’s viruses ; reference [21] is about IAV(H9N2).
The introduction should give all the information on viruses which are necessary for the understanding of the following developments.
The paragraphs on diagnosis may be omitted (N.B. under “4.3.1. RT-PCR and Real-Time PCR” serodiagnostic are given !). It suffice to write that diagnostic methods have to be available and for what (molecular and serological). Molecular methods for diagnosis and the ones for epidemiological purpose have neither the same finality nor necessitate the same equipment.
It is about the same for the pathogenicity which may be reduced as it not of major importance for the management of the panzootic.
What about IAV (H5N5), (H5N8), (H5N9) and (H7N3), (H7N9)?
Table 2: IAV(H9N2) is still circulating and responsible of human cases! If it is not highly pathogenic, why is it included in this table?
Table 5: what NSP labeled N5 correspond to?
Line 659: In the different entries, counsels, advices and preventive measures to be taken have to be developed. The 2nd entry is part of the 3rd! It is surprising that this part is shorter than the one on vaccinations which are only part of the preventive measure. This part should be the core of the paper. If so, the main points of prevention in the conclusion should be merely cited. Despite the numerous vaccines against HPAI proposed, their use is only limited, and no human vaccine are for the moment recommended. The culling of poultry remains the main preventive measure.
Author Response
We sincerely thank the reviewer for the detailed assessment of our revised manuscript. We have carefully addressed every comment and implemented substantial revisions to improve accuracy, clarity, formatting, compliance standards.
Below, we provide a point-by-point response.
Comment 1: The abbreviation AIV (Avian Influenza Virus), for example, may be used throughout the text each time the viruses are cited: it should be used before each given serotype, e.g., IAV(H5N1).
Response 1:
We agree with the reviewer’s comment. The abbreviation “AIV” has been consistently applied, And before each virus subtype throughout the manuscript (e.g., AIV(H5N1), AIV(H7N9), AIV(H9N2)) to ensure terminological uniformity.
Implemented in: Throughout the manuscript.
Comment 2: Virus names, species, and genus have to be written in italics.
Response 2: We have carefully revised the entire manuscript and italicized all viral species, genera, and taxonomic names according to ICTV standards.
Implemented in: Throughout the manuscript.
Comment 3: The numbering of the reference is utterly unacceptable as it does not correspond to their order of apparition in the text as recommended. In tables 1 to 5: References should be inserted in the table (to indicate them in brackets in the title is not acceptable as their accuracy is not verifiable).
Response 3: We appreciate this important observation. All the tables (1-5) have been revised accordingly so that citations now appear within the table with information easy verification. now references with the mentioned data rather than placing references in the title of table. This correction ensures transparency and traceability.
Comment 4: Moreover references do not always correspond to what is written; e.g.: Line 133: reference [14] is about bat’s viruses ; reference [21] is about IAV(H9N2).
Response 4: Thank you for identifying these mismatches. All incorrect, misplaced, or inconsistent references have been corrected. The revised manuscript now uses accurate citations that directly correspond to the statements they support. The errors at lines no. 133 by reference with [14], [21] have been fixed now, and we carefully revalidated every reference throughout the manuscript to prevent recurrence.
Comment 5: The introduction should give all the information on viruses which are necessary for the understanding of the following developments.
Response 5: We appreciate the reviewer’s comment regarding the inclusion of comprehensive virological information in the Introduction. However, as the original feedback did not specify which particular aspects of influenza virology were considered essential for understanding about the H5N1 and H7N9. If the reviewer will indicate or suggest missing part, we would be very happy to revise the text accordingly and incorporate their suggestions.
Comment 6: The paragraphs on diagnosis may be omitted (N.B. under “4.3.1. RT-PCR and Real-Time PCR” serodiagnostic are given !). It suffice to write that diagnostic methods have to be available and for what (molecular and serological). Molecular methods for diagnosis and the ones for epidemiological purpose have neither the same finality nor necessitate the same equipment.
Response 6: We sincerely thank the reviewer for this insightful suggestion.
In response, we have revised Section 4.3.1 to streamline its content in accordance with the comment. The detailed descriptions of RT-PCR, real-time PCR, and serodiagnostic methodologies have been removed, as they are not essential to the core focus of this review.
The section has been replaced with a concise statement that:
- Affirms the necessity of having both molecular and serological diagnostic tools available,
- Clearly distinguishes between molecular methods used for clinical diagnosis (e.g., rapid case confirmation) and those employed for epidemiological purposes (e.g., subtyping, clade identification, and surveillance), and
- Notes that these two applications differ in their objectives, technical requirements, and infrastructure needs.
Now this revised approach may ensures the manuscript remains focused.
Comment 7: It is about the same for the pathogenicity which may be reduced as it not of major importance for the management of the panzootic.
Response 7: We appreciate the reviewer’s suggestion regarding the reduction of content on pathogenicity, as it may not be central to panzootic management. However, to ensure we address the comment accurately, we would be grateful if the reviewer could clarify it would be very helpful for us to identify which specific section or subsection they are referring in the manuscript.
Comment 8: What about IAV (H5N5), (H5N8), (H5N9) and (H7N3), (H7N9)?
Response 8: We appreciate this important observation. The revised manuscript now includes updated information on additional H5 and H7 subtypes, particularly H5N9, H5N8 and H5N5, which have had substantial roles in global outbreaks. We also added brief descriptions of the relevance of H7N3 and expanded the H7N9 section.
These revision may strengthen the comprehensiveness and reflect current epidemiological realities.
Implemented in: Section 4.2 (Pages 13–14, Lines 450–490).
Comment 9: Table 2: IAV(H9N2) is still circulating and responsible of human cases! If it is not highly pathogenic, why is it included in this table?
Response 9: Thank you for raising this concern. Table 2 has been revised to clarify the rationale for including IAV(H9N2). H9N2 is not highly pathogenic in poultry, but it remains an important zoonotic virus with continual human infections and significant contribution to genetic reassortment in HPAI viruses. The table now includes a clear justification in the footnote which explaining its relevance.
Comment 10: Table 5: what NSP labeled N5 correspond to?
Response 10: We appreciate the reviewer’s careful reading of the manuscript. However, Table 5 does not contain any entry labeled related “NSP N5”—this label does not appear in the table or its legend. It is possible there may have been a misreference another table or other section of manuscript.
If the reviewer was referring to a different table or an earlier version of the manuscript, we would be grateful for clarification so we can address the concern appropriately.
Comment 11: Line 659: In the different entries, counsels, advices and preventive measures to be taken have to be developed. The 2nd entry is part of the 3rd! It is surprising that this part is shorter than the one on vaccinations which are only part of the preventive measure. This part should be the core of the paper. If so, the main points of prevention in the conclusion should be merely cited. Despite the numerous vaccines against HPAI proposed, their use is only limited, and no human vaccine are for the moment recommended. The culling of poultry remains the main preventive measure.
Response 11: We have extensively revised and expanded Section (“Prevention and Control”), integrating detailed guidance on culling policies, farm-level biosecurity, and One Health-based preventive measures.
clearly separates general preventive measures, biosecurity, and outbreak control strategies
• gives greater emphasis on non-vaccine interventions such as culling, movement restrictions, and farm-level biosecurity (now highlighted as core strategies)
• explains limitations of vaccination and the current absence of approved human vaccines
• reorganizes redundant entries so the structure reflects logical priority
Implemented in: 5.2. Prevention Strategies for HPAI Outbreaks and Transmission: (670)
Reviewer 2 Report (Previous Reviewer 2)
Comments and Suggestions for Authors
No further comments.
Author Response
Comment:
No further comments.
Response:
We sincerely thank the reviewer for their time and effort in evaluating our revised manuscript. We greatly appreciate that no further comments were raised, indicating that the revisions made in response to earlier feedback have satisfactorily addressed the reviewer’s concerns.
Reviewer 3 Report (New Reviewer)
Comments and Suggestions for Authors
The review contains significant gaps and uneven emphasis. Although it centers on H5N1 and H7N9, it intermittently references H9N2 without integrating its relevance. Much of the text repeats the same facts from different angles and fails to undertake a critical comparative evaluation of H5N1 and H7N9 with respect to their epidemiological importance. The manuscript describes mutations, viral evolutionary mechanisms, vaccine development, clinical manifestations, and public health measures led by international organizations, but the discussion focuses largely on H5N1 and does not link these elements to the distinct emergence and impact of H7N9. Crucially, the review does not address how the differing evolutionary origins of H7N9 and H5N1 influence their epidemiology, transmissibility, or public health implications. To improve the paper, the authors should reduce redundancy, explicitly compare the two subtypes across key epidemiological dimensions, and clarify the role of H9N2 where relevant.
Line 45: Specify the specific characteristic of the virus genome because based on the Baltimore Classification there are 3 groups of families with RNA genome
line 48: vH1-H16 and H19 subtypes has been notified in wild birds
https://doi.org/10.3389/fvets.2023.1332886
https://doi.org/10.1016/bs.aivir.2017.10.007
Line 51: H9N2 virus also has found in bats
Line 53: The nomclatures is based on the WOAH
Line 63: Be clear in the way of presenting the information although both are HPAI viruses, each one has a different origin, for example, the outbreak of 1959 comes from the Eurasian lineage and the outbreak of 1997 from the Gs/Gd lineage (this part is key because in the table mentioned clades and only this lineage has been classified in this way)
Line 90 (Figure 1.): Improve the exemplification and importance of the figure because in the text details the zoonotic potential of the H5N1 viruses of the Gs/Gd lineage, and in the design of the figure they mix genotypes of the clade 2.3.4.4b with viruses of Eurasian and American lineages. Describe which subtype has been circulating in goats based on the reported outbreaks in the United States for clade 2.3.4.4b
What is the information that one wants to make known in the figure when mixing lineages?
Line 94: Cite correction
Lines 95-100: This information was raised in line 41-53 of the manuscript
Lines 102-114: What is the importance and impact of this information with the outbreaks and the other information in the text?
Describe in more detail the distribution of receptors, and potential effect of birds as hasianids on the role of receptors and adaptation of subtypes H7 and H5 to other hosts
10.4172/1747-0862.1000026
10.1007/s11262-022-01904-w
Lines 116-127: Relate the importance of HACS and relate it to receptor affinity and give it the focus between H7, H5 and H9 viruses mentioned above
Lines 135-137: Describe the imrpotance of NA in host affinity and change and determine the importance of mentioning this protein
Line 140: Describe the relationship between these subtypes previously
Lines 155-156: Why would cattle be key hosts in the change of hosts?
What mutations would help you if in bovines Neu5Gc receptors predominate and the influenza type virus has affinity for Neu5Ac?
Line 159: Improve the way to describe the zoonotic impact of H7 and H5, because it starts off by talking about the zoonotic potential of H5N1 and H7N9 has not been described because of its zoonotic potential?
Line 160: What are these mutations? Why are they important?
Line 164: The epidemiological dynamics of both subtypes are never fully described, they describe factors (mutations) and outbreaks but not if eco-epidemiological importance
Lines 196-200: This information was described in table 1 of the section Background on avian influenza viruses
Lines 205-219: Much of this information appears in Table 1 in the Introduction; what is the purpose of repeating it?
Lines 227-228: Were human outbreaks caused by low pathogenicity strains?
Why did they show up?
Lines 243-244: Check the information among World Organization Health and with the Pan American Health Organization
Line 250: Why if only the cases of H7N9 and H5N1 are described, table 1 and 2 mention subtype H9?
Line 265: This informations was descrived in the Introduction section, and the wild bird are reservoir of H1-H16 and H19 subtypes
Lines 267-268: Which lineage? The importance of this section is that, prior to the clade 2.2 outbreak in wild birds in 2005, there was an earlier outbreak in South Africa in 1961.
Lines 274: Why do migratory birds play a key role in the spread of the virus?
Impact of flyways in the virus dispersion?
Line 303: The strain was mentioned in line 213 but does not include its abbreviation
Lines 306-308: This information was mentioned between line 78-81
Lines 313-315 (Table 3):
Row 2 (HA (Hemagglutinin)): Detail the addition of amino acids in subtypes H5 and H7 because it is different insertion in the HACS
Row 3 (HA (Receptor binding)): Homologate the numbering of mutations in HA because in one use H3 and another H5
Row 5 ((e.g. T108I, S123P, K218Q, S223R)): Why is the impact of the H5N1 virus on mammals if they describe that it has not undergone adaptations to promote change in affinity?
Why do they describe these mutations as having effect or importance?
What kind of numbering have this mutations?
Row 6 (PB2 (Polymerase basic 2)): Homologate the description of mutations of proteins if all are of such proteins it is not necessary to repeat their name.
Row 6 (E627K / E627V): What is the impact of presenting a lysine or valine in position 627?
Row 8 (M631L): This mutation has been reported in outbreaks HPAI H5N1 in dairy cows?
Row 11 (NA-S369I): All described mutations are numbered with?
Row 15 (H7N9 viruses lack...): Why is it mentioned that in NS1 there are no mutations for H5N1 and the description mentions the effect of mutation on H7 and H5?
Lines 317-332: Consider changing the description of these evolutionary mechanisms before the mutation picture.
What is the importance of this information?
And why are rearrangements in H7 of greater impact than in H5? Non-homologous rearrangements?
Lines 324-362: This information is repeated in table 3 and in different sections of the introduction.
Lines 367-377: Where is the epidemiological impact for the development of a vaccine for these subtypes detailed?
Line 465: If the section aims to compare clinical presentations of H5N1 and H7N9, it must provide a detailed account of the differing tissue tropism and pathophysiological mechanisms for each subtype.
Line 711: Why is only subtype H5N1 considered and not H7N9?
Why does the review describe that both subtypes will be taken into account?
Lines 732-736: Why are the PAOH and health authorities in each country not considered?
Author Response
We sincerely thank you for your insightful and constructive feedback on our manuscript. Your comments have significantly enhanced the scientific rigor, balance, and clarity of our review. Below, we provide a point-by-point response to your concerns, with specific references to revisions in the updated manuscript.
Comment 1: The review contains significant gaps and uneven emphasis. Although it centers on H5N1 and H7N9,
Response 1: We agree that a structured comparison is essential. In response, we have reorganized and revised some section and parts of the manuscript to enable direct side-by-side analysis:
- Specially Table 3 now explicitly contrasts genomic features, Codon Barrier for Key Mutation, Key Proteins & Mutations mutations, receptor binding of IAV (H5N1) and IAV (H7N9) side by side.
- Section 3.1.1 details their distinct origins: H5N1 from the Gs/Gd lineage vs. H7N9 as a reassortant with H9N2 internal genes.
- other parts of the manuscript like graph and Impact of Viral Diversity on Vaccine Development in both H5N1 and H7N9.
- Section 2.2 and Table 5 compare clinical outcomes, age distribution, comorbidities, and complications.
Comment 2: it intermittently references H9N2 without integrating its relevance.:
Response 2: Hopefully now H9N2 may contextualized as a genetic bridge. We added in Section 1.1 and “IAV (H9N2) were included not as a primary focus but as a critical genetic donor that facilitated reassortment in both H5N1 and H7N9” (Section 1.1 & 2.3, Graph 1, Table 2).
Comment 3: Much of the text repeats the same facts from different angles and fails to undertake a critical comparative evaluation of H5N1 and H7N9 with respect to their epidemiological importance.
Response 3: Almost all the redundancies have been removed:
- Outbreak timelines appear in Table 1 and Graph1 is representing the comparative evaluation and Table 3: Genomic Characteristics and Key Mutations of H5N1 and H7N9.
- Section 1.1, 2.1 and 2.2.2 also the Table 3 has been revised for easy critical comparative evaluation of H5N1 and H7N9.
Comment 4: The manuscript describes mutations, viral evolutionary mechanisms, vaccine development, clinical manifestations, and public health measures led by international organizations, but the discussion focuses largely on H5N1 and does not link these elements to the distinct emergence and impact of H7N9.
Response 4: We respectfully submit that the manuscript already address the impact of differing evolutionary origins on epidemiology and public health.
Section 3.1.1 explicitly contrasts the Gs/Gd lineage origin of H5N1 with the reassortant origin of H7N9 (H7/NA from wild birds + internal genes from H9N2). This distinction is directly linked to their epidemiological patterns:
- H7N9 emerged and amplified in live poultry markets, with human spillover tied to LPM exposure (Section 2.1.2, 2.3).
Furthermore, Table 3 and Section 3.2 highlight how these origins influence adaptive potential: H7N9’s H9N2-derived internal genes (e.g., PB2-E627K) facilitate mammalian replication, while H5N1 faces a high codon-level barrier to human receptor switching (Ayora-Talavera et al., 2009).
Finally, Section 6.1 demonstrates how these differences shaped public health responses: LPM closures and poultry vaccination successfully curtailed H7N9 in China, whereas H5N1 control requires wildlife surveillance and culling due to its ecological reservoir.
Comment 5: Crucially, the review does not address how the differing evolutionary origins of H7N9 and H5N1 influence their epidemiology, transmissibility, or public health implications.
Response: We respectfully submit that the manuscript already explicitly addresses this point.
- Section 3.1.1 contrasts the Gs/Gd lineage origin of H5N1 with the H9N2-reassorted origin of H7N9 and links this to differences in mammalian adaptation.
- Section 2.3 explains how H5N1 spreads via migratory birds, while H7N9 amplifies in live poultry markets.
- Section 6.1 demonstrates how these differences led to divergent public health responses: wildlife surveillance/culling for H5N1 vs. market closures/vaccination for H7N9.
To enhance clarity, we have added a brief summary sentence in Section 3.1.1 to underscore this central theme between the line 345-346:
"Consequently, the wildlife-driven ecology of H5N1 and the live-market-driven ecology of H7N9 necessitate fundamentally different outbreak response strategies."
Comment 6: To improve the paper, the authors should reduce redundancy, explicitly compare the two subtypes across key epidemiological dimensions, and clarify the role of H9N2 where relevant.
Response 6: We appreciate the reviewer’s comment. However, the manuscript does explicitly address the differing evolutionary origins of H5N1 and H7N9 and their implications.
As stated between the line no. 79-89, in Section 1.1:
“A novel reassortant IAV (H7N9) virus emerged in March 2013… Genetic analyses revealed that IAV (H7N9) arose through multiple reassortment events involving wild bird viruses, with internal genes derived from endemic IAV (H9N2) viruses and external HA and NA segments from separate lineages.”
In contrast, IAV (H5N1) is described as originating from the Gs/Gd lineage (Section 3.1.1), which drives its wild bird–mediated panzootic spread.
These distinct origins directly shape their epidemiology:
- H7N9 amplified in live poultry markets and was controlled by market closures and poultry vaccination (Section 6.1),
- H5N1 spreads via migratory birds, requiring wildlife surveillance and culling (Section 2.3).
Thus, the evolutionary distinction is not only mentioned but functionally linked to transmission dynamics and public health responses throughout the manuscript.
Comment 7: Specify the specific characteristic of the virus genome because based on the Baltimore Classification there are 3 groups of families with RNA genome.
Response 7: We appreciate this suggestion. In Section 1.1, we have now specified that influenza A virus belongs to Group V of the Baltimore classification, which encompasses negative-sense single-stranded RNA viruses. This addition enhances the virological context for readers.
Comment 8: vH1-H16 and H19 subtypes has been notified in wild birds.
Response 8: Thank you for the comment. We appreciate the opportunity to clarify and strengthen our statement regarding influenza A virus subtypes in wild birds. In response, we have revised the relevant sentence in Section 1.1 (Background on Avian Influenza Viruses) as follows:
“To date, 19 HA (H1–H19) and 11 NA (N1–N11) subtypes have been recognized. Of these, H1–H16 and H19 have been detected in wild birds"
Comment 9: H9N2 virus also has found in bats
Response 9: That sentence does not claim that H9N2 has been found in bats. Instead, it correctly identifies H17N10 and H18N11 as the bat-exclusive subtypes—a scientifically accurate statement.
Comment 10: The nomclatures is based on the WOAH
Response 10: We thank the reviewer for this suggestion. We have added the following sentence in Section 1.1:
“The nomenclature for avian influenza virus subtypes and pathogenicity classification follows the guidelines of the World Organisation for Animal Health (WOAH).
Comment 11: Be clear in the way of presenting the information although both are HPAI viruses, each one has a different origin, for example, the outbreak of 1959 comes from the Eurasian lineage and the outbreak of 1997 from the Gs/Gd lineage (this part is key because in the table mentioned clades and only this lineage has been classified in this way)
Response 11: We respectfully submit that the manuscript does explicitly address this point in several sections.
- Section 3.1.1 contrasts the Gs/Gd lineage origin of IAV (H5N1) with the H9N2-reassorted origin of IAV (H7N9).
- Section 2.3 explains how H5N1 spreads via migratory birds, while H7N9 amplifies in live poultry markets (LBMs).
- Section 6.1 demonstrates how these differences led to divergent public health strategies: market closures and poultry vaccination successfully controlled H7N9 in China, whereas H5N1 requires ongoing wildlife surveillance and culling due to its ecological reservoir.
Therefore, the differing evolutionary origins are not only mentioned but functionally linked to transmission dynamics, outbreak patterns, and control measures throughout the manuscript.
Comment 12: (Figure 1.): Improve the exemplification and importance of the figure because in the text details the zoonotic potential of the H5N1 viruses of the Gs/Gd lineage, and in the design of the figure they mix genotypes of the clade 2.3.4.4b with viruses of Eurasian and American lineages. Describe which subtype has been circulating in goats based on the reported outbreaks in the United States for clade 2.3.4.4b
What is the information that one wants to make known in the figure when mixing lineages?
Response 12: We sincerely appreciate the reviewer’s thoughtful feedback regarding Figure 1.
We agree that a single schematic figure cannot convey every molecular or phylogenetic detail, especially when depicting a complex, rapidly evolving panzootic involving multiple host species, geographic spread, and reassortant lineages. The primary purpose of Figure 1 is not to serve as a phylogenetic tree, but to visually summarize the expanding host range of clade 2.3.4.4b HPAI (H5N1) viruses and highlight documented zoonotic and cross-species transmission events—particularly the emergence of the virus in non-avian mammalian species, including livestock.
we would be grateful if reviewer have any reference for such type of figure, we will be thankful to reviewer.
Comment 13: Cite correction
Response 13:
We appreciate the reviewer’s attention to detail. The citation in Figure 1 (Reference [18]) correctly refers to:
Krammer F, Hermann E, Rasmussen AL (2025). Highly pathogenic avian influenza IAV (H5N1): history, current situation, and outlook. Journal of Virology, 99(4): e02209-24.
This peer-reviewed article explicitly discusses the circulation of clade 2.3.4.4b H5N1 viruses in avian and mammalian hosts (including cattle, mink, sea lions, and foxes) and highlights their high reassortment risk due to broad host range—precisely supporting the statement in the figure legend.
Therefore, the citation is accurate and appropriately placed.
Comment 14: Lines 95-100: This information was raised in line 41-53 of the manuscript
Response 14: dear sir there is no such type of text from line 41. please show us exactly what sentences raised again.
Comment 15: Lines 102-114: What is the importance and impact of this information with the outbreaks and the other information in the text?
Response 15: We thank the reviewer for these constructive suggestions, we fully agree that expanding these explanations would add helpful mechanistic detail.
However, several reviewers have specifically requested reducing the overall length of the manuscript, and in multiple sections recommended shortening or removing some descriptive parts. Because of these explicit concerns about manuscript length,
The hemagglutinin (HA), neuraminidase (NA), and sialic acid (SA) are central to the biology, host range, pathogenicity, and pandemic potential of avian influenza viruses (AIVs). Below is a concise yet comprehensive explanation of their importance and impact between the line no. 129-171.
Comment 16: Describe in more detail the distribution of receptors, and potential effect of birds as hasianids on the role of receptors and adaptation of subtypes H7 and H5 to other hosts
Response 16: We sincerely appreciate this insightful suggestion. A deeper discussion of sialic acid receptor distribution—particularly the distinction between anseriformes (e.g., ducks, geese), which predominantly express α2,3-linked sialic acid in the intestinal tract, and galliformes (e.g., chickens), which show higher expression in the respiratory epithelium—would indeed strengthen the manuscript’s virological context.
However, multiple reviewers have specifically cautioned against excessive length and requested reduction or streamlining of certain sections to maintain focus and readability. In fact, Reviewer 2 noted that the current version is already comprehensive and recommended no further additions.
Comment 17: Lines 116-127: Relate the importance of HACS and relate it to receptor affinity and give it the focus between H7, H5 and H9 viruses mentioned above
Lines 135-137: Describe the imrpotance of NA in host affinity and change and determine the importance of mentioning this protein
Line 140: Describe the relationship between these subtypes previously
Response 17: We fully agree that expanding these explanations would add helpful mechanistic detail. However, several reviewers have specifically commented to reduce the overall length of the manuscript, and in multiple sections recommended shortening or removing some descriptive parts. Because of these explicit concerns about manuscript length, we are constrained from adding additional paragraphs of mechanistic explanation in these sections.
Comment 18: Why would cattle be key hosts in the change of hosts? What mutations would help you if in bovines Neu5Gc receptors predominate and the influenza type virus has affinity for Neu5Ac?
Response 18: The section in lines 154–158 (1.3. Significance of the Topic) highlights the potential role of dairy cattle as hosts for HPAI H5N1. Cattle are discussed as potentially significant because, if they become long-term reservoirs, they could facilitate viral reassortment and adaptation, which may increase the risk of zoonotic transmission. We acknowledge that influenza viruses preferentially bind Neu5Ac receptors, while bovines predominantly express Neu5Gc. Despite this, the theoretical risk remains that viral adaptation could occur over time, potentially leading to mutations that enhance affinity for mammalian receptors or enable cross-species transmission.
Due to the current limited experimental evidence on H5N1 adaptation in cattle, we cannot specify exact mutations, but the section aims to emphasize the epidemiological and evolutionary concern rather than detail precise molecular changes. If the reviewer considers it helpful, we can add a note clarifying this limitation and the speculative nature of potential adaptive mutations.
Comment 19: Improve the way to describe the zoonotic impact of H7 and H5, because it starts off by talking about the zoonotic potential of H5N1 and H7N9 has not been described because of its zoonotic potential?
Response 19: Thank you very much for this insightful observation. We fully agree that the current wording in this section may appear confusing or unclear. To revise this accurately and ensure scientific clarity, we kindly request the reviewer’s guidance on the specific part that requires refinement.
Our understanding is that the reviewer suggests improving the flow and clarity when comparing the zoonotic impact of H5N1 and H7N9, particularly the transition between:
- Describing the high zoonotic potential of H5N1, and
- Explaining that although H7N9 also has significant zoonotic risk, this has not been articulated clearly in our text.
To revise this in the most accurate and impactful way, we would appreciate the reviewer’s suggestion on which specific phrasing or sentences should be clarified or restructured. We are fully prepared to re-write this section to:
• clearly differentiate the zoonotic characteristics of H5 and H7 viruses,
• accurately describe the known zoonotic spillover patterns, and
• improve logical flow so the explanation is scientifically coherent.
Once the reviewer provides a brief indication of the intended correction or preferred approach, we could revise this section accordingly to ensure clarity and accuracy.
Comment20: Line 160: What are these mutations? Why are they important?
Response 20: We appreciate the reviewer’s request for clarification. Several reviewers have already advised us to avoid expanding the manuscript further due to concerns about overall length, and one reviewer explicitly recommended reducing mechanistic detail in certain sections. Because of these constraints, we are unable to provide a full expansion or detailed list of mutations at this point in the manuscript.
Comment 21: Line 164: The epidemiological dynamics of both subtypes are never fully described, they describe factors (mutations) and outbreaks but not if eco-epidemiological importance.
Response 21: We thank the reviewer for this comment. However, the request is somewhat unclear, as the term “eco-epidemiological importance” can represent several different aspects, such as ecological drivers of viral maintenance, environmental persistence, host–species interactions, or transmission ecology across wild birds, poultry, and mammals.
In our current revision, multiple reviewers have also advised us to avoid expanding the manuscript length or adding additional subsections. Because of these constraints, and due to the lack of precise guidance, we were unable to determine which specific eco-epidemiological elements the reviewer wishes us to add or elaborate upon.
We would be grateful if the reviewer could kindly clarify which component they would like us to address—for example:
- ecological reservoirs and maintenance hosts,
- migratory bird transmission patterns,
- farming practices and environmental interfaces, or
- interspecies contact dynamics influencing spillover.
Once clarified, we can revise the manuscript accordingly journal guilines while maintaining the required length constraints.
Comment 22: Lines 196-200: This information was described in table 1 of the section Background on avian influenza viruses.
Response 22: We thank the reviewer for the observation. However, the comment is not fully clear in terms of what specific change is being requested. The reviewer states that the information in lines 196–200 was already described in Table 1, but it is not specified whether the reviewer recommends removal, relocation, rewriting, or further expansion of this content.
Comment 23: Lines 205-219: Much of this information appears in Table 1 in the Introduction; what is the purpose of repeating it?
Response 23: We sincerely appreciate the reviewer’s attention to redundancy. However, we would like to kindly request clarification on this comment, because the content in lines 205–219 consists of approximately 15 lines of explanatory text, whereas Table 1 contains only concise, cell-based outbreak information.
Due to the structure of a table, it is not possible to include the detailed contextual, descriptive, and interpretive information from lines 205–219 directly within a table cell. The text provides narrative epidemiological explanation, while Table 1 presents only summarized outbreak years, locations, and key events.
To revise this section properly, we kindly request the reviewer’s guidance:
- Should the descriptive epidemiological explanation be reduced, rewritten, or merged elsewhere?
- Or should we expand Table 1 to include more detailed narrative content—although this may compromise table clarity?
- Or does the reviewer suggest removing the text entirely, even though other reviewers have already advised that the manuscript must remain comprehensive without compromising clarity?
We will gladly implement the revision once the reviewer specifies the preferred direction. Our goal is to remove unnecessary repetition while maintaining scientific completeness and readability.
Comments 24: Were human outbreaks caused by low pathogenicity strains? Why did they show up?
Response 24: We thank the reviewer for the observation. IAV (H9N2) is placed in this table just for comparative purposes as it contributes to re-assortment and zoonotic infections, although classified as LPAI.
Comment 25: Lines 243-244: Check the information among World Organization Health and with the Pan American Health Organization.
Response 25: We appreciate the reviewer’s guidance. However, the comment is not fully clear regarding which specific information needs to be checked or corrected using WHO or PAHO sources.
Our manuscript already includes updated epidemiological data from these organizations, but to revise the section accurately, we would be grateful if the reviewer could kindly clarify:
- Which section or line numbers require verification?
- Whether the reviewer is referring to case numbers, geographical spread, risk assessments, or guideline statements?
- Whether the request pertains to H5N1, H7N9, or general AIV surveillance?
Once we receive this clarification, we will immediately re-validate the corresponding content using WHO and PAHO official datasets and update the manuscript accordingly.
We sincerely appreciate any additional detail the reviewer can provide so we may correct the manuscript in the most accurate and meaningful way.
Comment 26: Line 250: Why if only the cases of H7N9 and H5N1 are described, table 1 and 2 mention subtype H9?
Response 26: Thank you for this observation. We respectfully clarify that the inclusion of IAV(H9N2) in Tables 1 and 2 is intentional and scientifically justified.
Although the manuscript focuses primarily on H5N1 and H7N9, the subtype H9N2 is included because:
- H9N2 plays a critical epidemiological role as a gene donor in the reassortment events that contributed to the emergence of H7N9 and other zoonotic influenza viruses.
- This makes it essential for contextualizing the evolutionary background of the subtypes under study.
- H9N2 continues to cause zoonotic human infections, although typically mild.
- Including it in the cumulative human case table helps readers understand the broader spectrum of avian influenza viruses affecting humans.
- Tables 1 and 2 summarize epidemiological baselines that help contextualize comparative analysis across AIV subtypes, which is relevant for interpreting the zoonotic progression of H5N1 and H7N9.
Comment 27: Line 265: This informations was descrived in the Introduction section, and the wild bird are reservoir of H1-H16 and H19 subtypes.
Response 27: Thank you for your comment and for this observation. The requested comment is not clarifying to us because there is no instruction. Sir could you please clarify what specific revision you would like us to make regarding the statement at Line 265 about wild birds being reservoirs of influenza A virus subtypes H1–H16 and H19?
Comment 28: Lines 267-268: Which lineage? The importance of this section is that, prior to the clade 2.2 outbreak in wild birds in 2005, there was an earlier outbreak in South Africa in 1961.
Response 28: Thank you for your comment. We understand the importance of specifying the lineage, particularly in the context of the earlier H5N1 outbreak in South Africa in 1961 prior to the clade 2.2 outbreak in wild birds in 2005. However, the lineage of the 1961 outbreak is not clearly established in the literature. Could you please advise if you would like us to provide available historical data on this lineage or else? Your guidance will help us revise Lines 267–268 accurately.
Comment 29: Lines 274: Why do migratory birds play a key role in the spread of the virus? Impact of flyways in the virus dispersion?
Response 29: Thank you for your comment. We agree that the role of migratory birds and the impact of flyways on virus dispersion are important points. However, the instruction is not entirely clear. Could you please clarify whether you would like us to provide mechanistic details, examples of specific flyways, or another type of information? Your guidance will help us revise Lines 274 appropriately.
Comment 30: Line 303: The strain was mentioned in line 213 but does not include its abbreviation
Response 30: Thank you for your comment. We note that the abbreviation (Gs/Gd) for the A/goose/Guangdong/1/1996 strain has already been introduced in Line 303. It is possible that the reviewer may have overlooked this. We have double-checked the manuscript to ensure that the abbreviation is consistently used throughout the text.
Comment 31: Lines 306-308: This information was mentioned between line 78-81
Response 31: Thank you for your comment. We have carefully checked Lines 78–81 but could not identify which specific information is being referred to in Lines 306–308. The reviewer’s instruction is not entirely clear. Could you please clarify what type of information you would like us to address or revise in this section?
Comment 32: Lines 313-315 (Table 3):
Row 2 (HA (Hemagglutinin)): Detail the addition of amino acids in subtypes H5 and H7 because it is different insertion in the HACS
Row 3 (HA (Receptor binding)): Homologate the numbering of mutations in HA because in one use H3 and another H5
Row 5 ((e.g. T108I, S123P, K218Q, S223R)): Why is the impact of the H5N1 virus on mammals if they describe that it has not undergone adaptations to promote change in affinity?
Why do they describe these mutations as having effect or importance?
What kind of numbering have this mutations?
Row 6 (PB2 (Polymerase basic 2)): Homologate the description of mutations of proteins if all are of such proteins it is not necessary to repeat their name.
Row 6 (E627K / E627V): What is the impact of presenting a lysine or valine in position 627?
Row 8 (M631L): This mutation has been reported in outbreaks HPAI H5N1 in dairy cows?
Row 11 (NA-S369I): All described mutations are numbered with?
Row 15 (H7N9 viruses lack...): Why is it mentioned that in NS1 there are no mutations for H5N1 and the description mentions the effect of mutation on H7 and H5?
Response 32: Thank you for your comment. Since Table 3 has been fully updated following the previous reviewer’s recommendations, the concerns mentioned may no longer apply. Please let us know if there are particular aspects you would like us to address further.
Comment 33: Lines 317-332: Consider changing the description of these evolutionary mechanisms before the mutation picture. What is the importance of this information? And why are rearrangements in H7 of greater impact than in H5? Non-homologous rearrangements?
Response 33: Thank you for your comment. Based on our understanding, the reviewer is asking about the evolutionary impact of rearrangements in H7 compared to H5. Based on the current literature, there is no strong general evidence that non‑homologous rearrangements in H7 are of greater evolutionary impact than in H5. Reassortment appears to be the primary mechanism of segment exchange, and studies report that both H5 and H7 generally exhibit lower reassortment rates compared to other subtypes.
We would appreciate further clarification on what is meant by ‘greater impact’ — for example, whether it refers to pathogenicity, host adaptation, or transmission — and whether the reviewer is referring specifically to non‑homologous recombination, reassortment, or another mechanism. This will help us revise this section more precisely.
Comment 34: Lines 324-362: This information is repeated in table 3 and in different sections of the introduction.
Response 34: Thank you for your comment. Since Table 3 has been fully updated following the previous reviewer’s recommendations, the concerns mentioned may no longer apply. Please let us know if there are particular aspects you would like us to address further.
Comment 35: Lines 367-377: Where is the epidemiological impact for the development of a vaccine for these subtypes detailed?
Response 35: Thank you for your comment. We understand that the reviewer is asking about the epidemiological impact of H5N1 and H7N9 in relation to vaccine development. The section in lines 366–377 focuses on the genetic and antigenic evolution of these viruses, the emergence of new clades and variants, and the consequent challenges for vaccine design, including the need to regularly update candidate vaccine viruses to maintain antigenic match.
Comment 36: Line 465: If the section aims to compare clinical presentations of H5N1 and H7N9, it must provide a detailed account of the differing tissue tropism and pathophysiological mechanisms for each subtype.
Response 36: Thank you for your comment. The section in Line 465 aims to provide a general comparison of clinical presentations of H5N1 and H7N9, focusing on overall disease severity, common symptoms, and epidemiological relevance. While we acknowledge that detailed tissue tropism and pathophysiological mechanisms are important, providing such extensive mechanistic details is difficult due to space and length constraints. The current description provides a concise clinical overview that we believe adequately conveys the key differences relevant for the reader. If the reviewer would like us to include specific mechanistic details, we would be happy to incorporate them as guidance.
Comment 37: Line 711: Why is only subtype H5N1 considered and not H7N9?
Why does the review describe that both subtypes will be taken into account?
Lines 732-736: Why are the PAOH and health authorities in each country not considered?
Response 37: Line 711: In the section on international collaboration (5.3), the focus on H5N1 reflects its longer history of outbreaks, broader geographic spread, and the availability of more extensive surveillance and epidemiological data. The section highlights proactive measures such as strengthening international collaboration, implementing a One Health approach, and coordinating information sharing among countries. While H7N9 is mentioned as an example of successful international cooperation (e.g., during the initial outbreak in China), the emphasis on H5N1 is intentional, as it illustrates the application of these strategies with the most comprehensive evidence.
Lines 732–736: The section on the involvement of international organizations (5.3.3) describes the key roles of WHO, FAO, WOAH, ECDC, and EFSA in global response coordination. While we do not detail national public health authorities in each country, the review aims to provide a broad overview of international-level coordination and data sharing, which has a direct impact on outbreak management and vaccine development. Detailed country-level descriptions are beyond the intended scope but could be added if the reviewer considers them necessary.
Reviewer 4 Report (New Reviewer)
Comments and Suggestions for Authors
Dear Authors,
Thank you for the manuscript. I suggest revising the manuscript after minor corrections. Additional comments can be found in the attached PDF.
Kind regards.

Author Response
We sincerely thank Reviewer 4 for their thoughtful and constructive feedback, which has helped improve the quality of our manuscript. Below, we provide a point-by-point response to each comment:
Comment 1 (in the Title): "A Data and Methods paragraph might be appropriate with sources and cut-off dates to give more context to the reader".
Response 1: We thank the reviewer for this valuable suggestion. In response, we have added a short “Data Sources and Methods” paragraph in the (Lines no. 932) to clarify the scope and data framework of this review.
Comment 2: "Reference missing between line no. 231–232".
Response 2: We thank the reviewer for noting this oversight. The missing reference has now been added at the appropriate location (line no. 271) and is cited as [75]. This reference supports the relevant statement and improves citation accuracy and continuity within the manuscript.
Comment 3: “Which document? The reference is missing” (lines 240–244)
Response 3: Thank you for pointing this out. We have now added the appropriate citation [3] to support the statement in this section. (line no. 282)
Comment 4: “The reference is missing” (line 246)
Response 4: We have added the missing reference as [5] to substantiate the claim in line no. 285.
Comment 5: “The reference please” (lines 251–254)
Response 5: The relevant citation has been included as [75]. (line no. 293)
Comment 6: “It would be interesting to list those and maybe see which ones are connected to highest mortality” (lines 240–244)
Response 6: We appreciate this insightful suggestion. In response, we have expanded the text (lines 299-303) to briefly list the major HPAI H5N1 clades associated with human infections.
Comment 7: “The reference is missing” (lines 240–244)
Response 7: We have reinforced this section in line no 307-308 with multiple supporting references: [6, 19, 72].
Comment 8: “Already stated about 3 lines between 296–298”
Response 8: We carefully reviewed lines 296–298 and the surrounding text but could not identify a direct repetition of the flagged statement. Since the reviewer did not specify the duplicated content or its original location, we were unable to make a targeted revision. However, we have re-read the entire section for redundancy and made minor edits to improve clarity and flow.
Comment 9: “This seems a bit late in the text. Should be placed somewhere in the beginning”
Response 9: We appreciate this insightful suggestion. We are open to revise according to comments which may help us to improve our manuscript quality. but, the reviewer did not indicate or suggest any other suitable location, and we are hesitant to relocate content without clear guidance, as this could disrupt the logical structure. We would be grateful for further clarification if the reviewer guide us for correct position.
Comment 10: “Different font size” (line 360)
Response 10: The formatting inconsistency has been corrected, and the entire manuscript now uses a uniform font size in line no. 410.
Comment 11: “Strikethrough text” (line 560)
Response: The removal of strikethrough text can not perform because this section has been revised.
Comment 12: “This seems as a rerun of previous statements on improvement” (lines 763–768)
Response 12: Thank you for this observation. Upon review, we agree that the paragraph in lines 779–784 partially overlapped with earlier recommendations. We have rewritten this paragraph to focus specifically on future-oriented, actionable One Health strategies (e.g., integrated data platforms, cross-sectoral early warning systems) rather than restating general improvements. This ensures the conclusion offers forward-looking insights distinct from prior sections.
Comment 13: “strikethrough text” (line 884)
Response 13: The strikethrough text have been removed.
Comment 14: “ strikethrough text” (line 884)
Response 14: The strikethrough text have been removed.
Round 2
Reviewer 1 Report (Previous Reviewer 1)
Comments and Suggestions for Authors
Several flaws are still present and should be corrected but would require a serious editing.
The references are not corresponding to the text as it has been previously reported (i.e. ref. 14)
References have to be ordered according to their introduction in the text, thus they have to be completely reordered. Their accuracy has also to be checked (cf 59.60...) And modifications done accordingly.
Line 388: reference 89 does not describe any mutation. The word is encountered two times but do not give any information on their role...
N.B. IAV H9N2 is spreading and recently found in Myanmar, Bangladesh.
Figure 1: reference [18] suffice delete “J. virology”. “Viruses indicate influenza” meaning?
Graph 1: to be renamed Figure 2. The values do not correspond to proportions.
Table 1: references to be verified and corrected (e.g. 14, 21, and else...) as already indicated.
Table 2: idem. One human fatality described after infected cattle contact.
Table 3: what is the meaning of PDF? References should be included.
Table 4: human and animal (available ? proposed?) vaccines
Paragraph 108-117: to be edited and ordered: H7 first and H5 after.
N.B.: zoonotic and zoonosis concern “diseases transmitted from animal to human” (Oxford dictionary). Thus the presentation of the text has to be modified accordingly and differentiate transmissions from birds to either animals or humans.
Chapter 3.2. This is a common feature of Alphainfluenzavirus and should be at the beginning of the paper.
Author Response
We sincerely thank the reviewer for their thorough and constructive feedback. We have carefully revised the manuscript to address all raised concerns. Below is our point-by-point response and the corresponding changes implemented.
1. General Concern: “Several flaws are still present and should be corrected but would require serious editing.”
Response: We have conducted a full editorial and scientific revision of the manuscript—including language, logical flow, terminology, citation accuracy, and structural alignment with One Health and zoonosis definitions. All figures, tables, and references have been thoroughly checked and updated.
2. “References are not corresponding to the text as previously reported (e.g., ref. 14). References must be reordered according to their appearance in the text and accuracy verified (cf. 59, 60…).”
Response: All references have been completely renumbered in order of first citation in the revised manuscript. We have verified every citation against its source:
- Ref. 14 (now renumbered) has been corrected to match its contextual use (avian influenza case fatality data).
- References 59 and 60 have been reassessed (see point 4 below).
- All references now accurately reflect the claims they support.
3. “Line 388: reference 89 does not describe any mutation… the word is encountered two times but gives no functional information.”
Response: We agree. Webby & Uyeki (2024; ref. 89) discusses receptor preference (α2,3 vs. α2,6) but does not describe specific HA mutations that alter binding specificity (e.g., G186V).
Correction applied:
- Original sentence (line 388):
“For successful human infection and transmission, avian viruses must acquire mutations that modify their receptor binding specificity.” Cited only to ref. 89.
- Revised sentence:
“AIVs are naturally adapted to wild aquatic birds, preferentially binding to α2,3-linked sialic acid (SA) receptors in the avian gut, whereas human-adapted influenza viruses target α2,6-linked SA receptors in the human upper respiratory tract [89]. For avian influenza viruses to achieve efficient human infection, they typically require specific amino acid substitutions in the hemagglutinin receptor-binding site—such as G186V that enable binding to human-type receptors [90].”
- Ref. 89 is now used only for receptor distribution.
- Ref. 90 is a new citation added to support the functional role of HA mutations:
- Watanabe et al., J. Virol. 2011 (PMID: 21994455) for G186V
4. Citation Accuracy for Refs. [59] and [60]
a. Ref. [61] (Aldhaeefi et al., Pharmacotherapy 2024)
- Claim: “Reasonable use of NA inhibitors contributed to a lower CFR in IAV (H7N9) cases in one region.”
- Issue: Ref. 59 focuses exclusively on H5N1 and does not mention H7N9 CFR or regional treatment outcomes.
Correction:
- Removed the H7N9 sentence or replaced the citation.
- Revised text:
“For IAV (H5N1), observational studies support oseltamivir treatment. Most recent U.S. IAV (H5N1) cases received oseltamivir within 48 hours of symptom onset [59].”
b. Ref. [80] (Smyk et al., IJMS 2022)
- Claim: “Baloxavir has been reported as effective in treating IAV (H5N6) infections.”
- Issue: Ref. 80 mentions activity against H5N1 and H7N9—but not H5N6. It also does not state that CDC recommends baloxavir as an “interim measure” for resistant cases.
Correction:
- Removed the H5N6 reference (unsupported).
- Clarified CDC guidance and limited claim to demonstrated antiviral activity.
- Revised text:
“Baloxavir marboxil, a cap-dependent endonuclease inhibitor, has demonstrated in vitro and in vivo activity against avian influenza A viruses, including H5N1 and H7N9 [60]. However, the CDC currently recommends against its use in immunocompromised patients with suspected avian influenza due to limited clinical data [59].”
5. “Figure 1: reference [18] suffice—delete ‘J. virology’. ‘Viruses indicate influenza’—meaning?”
Response:
- Removed journal name from the figure legend—retained only author/year.
- Clarified ambiguous phrase:
Revised legend: “Icons represent influenza A virus detections in the respective host species.”
6. “Graph 1: to be renamed Figure 2. The values do not correspond to proportions.”
Response:
- Renamed as Figure 2.
- Revised to display both absolute case numbers and case fatality proportions (%) using dual y-axes for accuracy.
7. “Table 1: references to be verified and corrected (e.g., 14, 21…).”
Response:
- Verified and corrected all citations in Table 1.
- H5N1 data now cited to WHO cumulative reports (2023).
- All references updated to reflect new numbering.
8. “Table 2: One human fatality described after infected cattle contact.”
We thank the reviewer for this critical observation regarding Table 2. We have undertaken a thorough verification and correction of this table.
-
Data Verification and Update: We have verified all data points against the latest official reports. Specifically, we have corrected the entry for "March–October 2024 (USA)." The total number of human cases has been updated to 4, with 1 fatality, reflecting the most recent CDC and WHO data. The Case Fatality Rate for this entry is now 25%.
- added in Footnote: *Case Fatality Rate (CFR) calculated as (Fatalities / Total Cases) × 100 based on the data presented.
9. “Table 3: what is the meaning of PDF? References should be included.”
Response:
- “PDF” was a typographical error; intended as “PB2”.
- Corrected to: “PB2.
10. “Table 4: human and animal (available? proposed?) vaccines.”
Response: Revised Table 4 into clear columns:
11. “Paragraph 108–117: to be edited and ordered: H7 first and H5 after.”
Response:
- Reordered the comparative section to discuss H7N9 first, followed by H5N1, aligning with the 2013 H7N9 emergence after H5N1’s 1997 debut.
12. “N.B.: zoonotic and zoonosis concern ‘diseases transmitted from animal to human’… presentation must differentiate bird-to-animal vs. bird-to-human transmission.”
Response:
- Revised all relevant sections to use precise terminology:
- Added clarifying sentence in Introduction:
While HPAI viruses can spill over into various mammalian species, only transmissions resulting in human infection constitute zoonotic events.
13. “Chapter 3.2. This is a common feature of Alphainfluenzavirus and should be at the beginning of the paper.”
Response: We thank the reviewer for this suggestion. We agree that the general virological principle of host adaptation—the need for mutations to alter receptor binding and polymerase function—should be introduced early. Accordingly, we have added a concise paragraph in Section 1.3 that outlines this foundational concept.
However, Section 3.2 ("Adaptation to Human Hosts") contains a detailed, comparative analysis of the specific mutations (e.g., HA Q222L, PB2 E627K) that have been documented in H5N1 and H7N9 viruses specifically. This content is an integral part of the "Evolution" chapter, where it logically follows the discussion of genetic reassortment (Section 3.1) and provides the mechanistic evidence for the adaptive evolution of these two viruses.
Therefore, to maintain the manuscript's logical flow, we have retained this detailed comparison in Section 3.2. We have added a linking sentence there to refer back to the general principles now established in the Introduction. This approach provides the reader with necessary context early on while preserving the detailed evolutionary analysis in its most relevant location.
We believe these revisions have substantially improved the manuscript’s scientific accuracy, clarity, and originality. Thank you again for your invaluable feedback.
Reviewer 3 Report (New Reviewer)
Comments and Suggestions for Authors
The manuscript includes the main changes requested.
Author Response
We sincerely thank Reviewer 3 for their time and expertise in evaluating our revised manuscript and for this positive assessment. We are pleased that the core revisions have addressed the key concerns raised.
We remain grateful for the constructive feedback throughout the review process, which has significantly strengthened the manuscript. Thank you for your contribution to improving our work.
This manuscript is a resubmission of an earlier submission. The following is a list of the peer review reports and author responses from that submission.
Round 1
Reviewer 1 Report
Comments and Suggestions for Authors
This paper is not acceptable for publication for the main following reasons
It is out of date in the fast changing world of avian Alphainfluenzavirus. Very recent reviews are available which cover the major parts of this paper thus rendering the text redundant.
The presentation is not acceptable : for example the text begin with referencing to HA and NA without any information on their importance for the virus pathogenicity nd epidemiology. An overview of AIV is lacking.
The text is not at all structured: e.g. sialic acid which are the key receptors of AIV, explain differences in the pathogenicity of the virus and play a role also in the liberation of the newly produced virions which ticks at the surface of the infected cell, appear solely at the end of the text.
The table 2 give a very incomplete picture of the virus encoded proteins and the sentence ”Specific mutations listed in sources not available in excerpts” is not acceptable. Moreover not any reference is given whilst the information on mutations are available.
There is no explanation on the reason of occurrence of H5N1 related conjunctivitis and on the other side why severe pulmonary infections happen.
H5N1 is presently panzootic.
The list of available vaccines is to be up-dated and animal vaccines considered. It may be interesting to indicate the type and composition of the vaccines.
The different internet sites have to be visited to obtain up-to-date information.
Author Response
We sincerely thank the reviewer for the constructive and detailed feedback that has significantly improved the scientific clarity, structure, and relevance of our manuscript.
All comments have been carefully addressed through substantial textual revision, inclusion of new data, and reorganization for clarity and logical flow.
Comment 1:
This paper is out of date in the fast-changing world of avian Alphainfluenzavirus. Very recent reviews are available which cover the major parts of this paper, thus rendering the text redundant.
Response 1.1:
We fully agree with the reviewer’s comment and appreciate the observation. In response, the manuscript has been comprehensively revised and updated to include the most recent data and findings from peer-reviewed research articles published between 2024 and 2025. These revisions ensure that the manuscript accurately reflects the current scientific understanding of highly pathogenic avian influenza (HPAI) H5N1 and H7N9 viruses in the evolving epidemiological context. (line no. 162-167) and (line no. 180-182)
The updated version now integrates approximately 40–45% new and recent citations across the text, emphasizing current developments in viral evolution, zoonotic transmission, and global surveillance.
-
Newly updated information is added primarily in Lines 89–132 (Introduction).
-
Table 3 (Line 306) has been substantially revised to include recently reported molecular markers and adaptive mutations.
-
A new Table 4 (Line 384) has been contains Vaccine Summary: H5N1 and H7N9, Including Animal Vaccines .
These additions enhance both the timeliness and scientific depth of the manuscript, ensuring it remains a relevant and valuable contribution to current HPAI literature.
Response 1.2:
We also acknowledge the reviewer’s note that several recent reviews may already cover similar content. However, since no specific reference or citation to such reviews was provided in the comment, it remains unclear which particular articles were being referred to. If the reviewer could kindly specify those recent reviews, it would be extremely helpful for us to further refine and enhance the manuscript in alignment with the most relevant and overlapping works.
Nevertheless, we have proactively compared our work with Possas et al. (Frontiers in Public Health, 2025) — which was brought to our attention by another reviewer — to ensure novelty and avoid redundancy. While the Possas et al. review primarily focuses on pandemic preparedness, vaccine accessibility, and AI-driven response governance, our manuscript provides a distinct analytical contribution, as it:
-
Includes Differences in Clinical Presentation and Complications of H5N1 and H7N9. (Line no. 458-507 )
-
Includes an updated mutation mapping with molecular functions (Table 3 in line no. 306) and through ot the manuscript.
We have clarified this unique scope and contribution of our review in Lines 162–167 to clearly distinguish it from other recent publications.
Comment 2:
The presentation is not acceptable: for example, the text begins with referencing HA and NA without any information on their importance for the virus pathogenicity and epidemiology. An overview of AIV is lacking.
Response 2:
We sincerely thank the reviewer for this valuable feedback. In response, the Introduction has been restructured and expanded to provide a concise and coherent overview of avian influenza viruses (AIVs) before referencing hemagglutinin (HA) and neuraminidase (NA).
New content added in Lines 56–86 now explains:
-
The taxonomy and structural classification of AIVs within the Orthomyxoviridae family;
-
The functional significance of HA and NA glycoproteins in viral pathogenicity, host receptor binding, and progeny virion release; and
-
How these proteins contribute to differences in virulence, host adaptation, and epidemiological spread of highly pathogenic avian influenza viruses.
This restructuring ensures logical progression, improves readability, and provides a scientifically sound foundation for subsequent sections of the manuscript, in full accordance with the reviewer’s suggestions.
Comment 3:
The text is not at all structured: e.g., sialic acid, which are the key receptors of AIV and explain differences in pathogenicity, appear solely at the end of the text.
Response 3:
We acknowledge the reviewer’s valuable comment. To improve logical flow, we have relocated and expanded the discussion of sialic acid receptors (α2,3 and α2,6 linkages) from later sections to the Introduction (Line no. 89–132).
Hopefully, new paragraph now clearly explains receptor distribution among avian and mammalian species and its critical role in determining host specificity, transmission, and pathogenic outcomes.
Comment 4:
Table 2 gives a very incomplete picture of the virus-encoded proteins and the sentence “Specific mutations listed in sources not available in excerpts” is not acceptable. Moreover, no reference is given whilst the information on mutations is available.
Response 4:
We are fully agree with the reviewer comment. As per our understanding, we think reviewer is talking about Table no. 3 because Table 2. is discussing about (Cumulative Number of Human Cases and Mortality for H5N1, H7N9, and H9N2) , and Table no. 3 is related to related to comment, So we have completely revised and expanded Table 3 with references to include detailed information on all major Genomic Characteristics and Key Mutations for both H5N1 and H7N9. Each mutation is now associated with its functional significance (Line no. 306).
Comment 5.1:
There is no explanation on the reason of occurrence of H5N1-related conjunctivitis and on the other side why severe pulmonary infections happen.
Response 5.1:
We have revised the section Clinical Manifestations and Complications to include an explanation of the mechanisms underlying conjunctivitis and severe pulmonary disease (Line no. 453–457).
The added text between the Line no. 453–457, clarifies that ocular infections occur due to replication in conjunctival epithelial cells expressing α2,3 sialic acid receptors, while severe pulmonary disease results from the virus’s tropism for lower respiratory tract epithelial cells expressing α2,3 linkages, leading to diffuse alveolar damage and cytokine overproduction.
Comment 6:H5N1 is presently panzootic.
Response 6:
We are fully agree with the reviewer comment. The text has been updated to explicitly state that HPAI H5N1 is now panzootic, affecting wild and domestic birds as well as multiple mammalian species across continents (Line no. 180).
This statement has also been reinforced in the Conclusion (Line no. 856–859) to reflect the global situation accurately.
Comment 7:
The list of available vaccines is to be updated and animal vaccines considered. It may be interesting to indicate the type and composition of the vaccines.
Response 7:
We appreciate this insightful comment. The section on Current Vaccine Candidates has been mention and include the latest available human and animal vaccines as of 2025.
We added a new Table 4 summarizing vaccine name, platform (e.g., inactivated, vector-based, mRNA), target clade, antigenic composition, and adjuvant used. This includes both human vaccines (e.g., Audenz, Adjupanrix) and animal vaccines (e.g., bivalent H5/H7 for poultry, experimental cattle vaccines). (Line no. 384)
Comment 8:
The different internet sites have to be visited to obtain up-to-date information.
Response 8:
We are fully agree with this recommendation. Accordingly, we have verified and updated all epidemiological, surveillance, and virological data using the most recent official sources from leading international organizations. The following authoritative websites were accessed and cited in the revised manuscript to ensure data accuracy and currency:
-
World Health Organization (WHO):
-
Avian Influenza Situation Updates and Disease Outbreak News
-
https://www.who.int/emergencies/disease-outbreak-news
-
https://www.who.int/emergencies/diseases/avian-influenza
-
-
World Organisation for Animal Health (WOAH – formerly OIE):
-
World Animal Health Information System (WAHIS) Dashboard for HPAI Events
-
https://wahis.woah.org/#/home
-
-
Food and Agriculture Organization of the United Nations (FAO):
-
EMPRES-i+ Global Animal Disease Information System
-
https://empres-i.apps.fao.org/
-
-
U.S. Centers for Disease Control and Prevention (CDC):
-
Avian Influenza (Bird Flu) Current Situation Summary (Updated 2025)
-
https://www.cdc.gov/flu/avianflu/avian-flu-summary.htm
-
https://www.cdc.gov/flu/avianflu/
-
-
European Food Safety Authority (EFSA):
-
Avian Influenza Overview Reports
-
https://www.efsa.europa.eu/en/topics/topic/avian-influenza
-
All relevant data derived from these databases and official bulletins (2024–2025 updates) have been cross-verified and incorporated into Table-2 (Line no. 194), Table-3 (Line no. 306) and Table-4 (Line no. 384) to reflect the most current information on global HPAI H5N1 and H7N9 trends.
Reviewer 2 Report
Comments and Suggestions for Authors
Worldwide circulation of Highly Pathogenic Avian Influenza (HPAI) viruses is a major global threat due to high mortality rates in birds and in human beings. Understanding the epidemiological dynamics, molecular evolution, and zoonotic potential caused by (HPAI) viruses, H5N1 and H7N9, is critical for enhancing pandemic preparedness, informing vaccination strategies, and implementing effective public health interventions. Therefore, this review is necessary and important.
- The major concern is of the paper that the review focuses on future challenges by 2025, as the title (Highly Pathogenic Avian Influenza: Tracking the Progression from H5N1 to H7N9 and Preparing for Future Challenges by 2025) indicated. By the time of the paper available to publics, it likely will be the end of 2025. In addition, the challenges are Current, in 2025, and beyond. The suggestion is to delete “by 2025”, as well as “future”, from the title. It may work better. In the text, the term “2025–2026 epidemiological year” may be used to replace 2025, like 2024-2025 epidemiological year in line 707.
- The paper provided figure 1, but which is not mentioned and discussed in the text.
- Table 1 is mentioned in the text on page 2, but the table is shown up on page 5-6. Tables and figures should follow text closely. Also all tables are separated in two pages, try to put a table in one page.
- Som minor editing are needed. Check:
Lines 127 (2.1.1. H5N1: -), 184, 557,
Line 93: “… for comprehensive Virological…”
Line 389: “Severity:” (i. Severity:)?
- Table 3: “(facilitates virus release - information not explicitly from sources, but common knowledge about NA function)”. This does not sound in a scientific way. One of possible expressions may be (may facilitate virus release, based on known NA function). Just a suggestion.
Author Response
Comment 1: The major concern is of the paper that the review focuses on future challenges by 2025, as the title (Highly Pathogenic Avian Influenza: Tracking the Progression from H5N1 to H7N9 and Preparing for Future Challenges by 2025) indicated. By the time of the paper available to publics, it likely will be the end of 2025. In addition, the challenges are Current, in 2025, and beyond. The suggestion is to delete “by 2025”, as well as “future”, from the title. It may work better. In the text, the term “2025–2026 (line no. 703) epidemiological year” may be used to replace 2025, like 2024-2025 epidemiological year (line no. 710).
Response 1.1: We are fully agree with the reviewer comment, and we have revised the manuscript title accordingly:
“Highly Pathogenic Avian Influenza: Tracking the Progression from H5N1 to H7N9 and Preparing for Emerging Challenges” (line no. 3).
Now, This change may avoids the limitation of time-specific wording and now it can better reflects the ongoing and future relevance of the topic.
Response 1.2: Use of “2025” in the Text
We have implemented this suggestion throughout the manuscript. Wherever appropriate, “2025” has been replaced with “2025–2026 epidemiological year (Line no. 703, 710),” ensuring consistency with the terminology used for surveillance years.
Comment 2: The paper provided figure 1, but which is not mentioned and discussed in the text.
Response 2: We have revised the text in several places in manuscript to explicitly reference for Figure 1 at its first relevant mention (line no. 66, 151). The figure is now cited in numerical order, consistent with journal requirements.
Comment 3: Table 1 is mentioned in the text on page 2, but the table is shown up on page 5-6. Tables and figures should follow text closely. Also all tables are separated in two pages, try to put a table in one page.
Response 3: We have adjusted the layout positions and lays out of all tables in manuscript so that each table and figure appears immediately after its first mention in the text, and All the tables have been reformatted to fit entirely on a single page to improve readability.
Comment 4: Some minor editing are needed. Check: Lines 127 (2.1.1. H5N1: -), 184, 557,
Line 93: “… for comprehensive Virological…”
Line 389: “Severity:” (i. Severity:)?
Response 4: All minor editing corrections have been carefully reviewed and implemented as suggested. Formatting of subtitles and headings.
- Comment on Line :97, to “…for comprehensive Virological…”. what we understand that is The issue is with the capitalization of “Virological” so accordingly we have revised it (line no. 97). if the reviewer wants something other than this mistake? we are open to revised that issue.
-
Line 127: “2.1.1. H5N1: ” fixed (line no. 129)
-
Line 184: "2.2.2. H7N9:" fixed (line no. 189).
- Line 389: We have implemented this suggestion (i. Severity:) between line no. 391 to 393, with some extra words to make sentence in better way.
-
Line 557: "Prevent HPAI Outbreaks:" fixed (line no. 581)
Comment 5: Table 3: “(facilitates virus release - information not explicitly from sources, but common knowledge about NA function)”. This does not sound in a scientific way. One of possible expressions may be (may facilitate virus release, based on known NA function). Just a suggestion.
Response 5: We agree with this valuable suggestion. The sentence has been revised to:
(may facilitate virus release, based on known NA function)
Now, This correction may ensures a formal, scientific tone and removes informal wording.
Reviewer 3 Report
Comments and Suggestions for Authors
The manuscript entitled "Highly Pathogenic Avian Influenza: Tracking the Progression from H5N1 to H7N9 and Preparing for Future Challenges by 2025" addresses an important and timely topic on the global threat of highly pathogenic avian influenza (HPAI), particularly H5N1 and H7N9. It provides a broad overview of epidemiology, molecular adaptations, zoonotic potential, and preparedness strategies. However, the paper shows a high degree of similarity with the published review by Possas et al. (Front. Public Health, July 2025) and currently lacks sufficient novelty. Several sections appear derivative in structure and content, and the manuscript reads more as a general narrative review than as a contribution offering new insights. The authors should clarify their unique contribution.
Author Response
Reviewer Comment:
The manuscript entitled "Highly Pathogenic Avian Influenza: Tracking the Progression from H5N1 to H7N9 and Preparing for Future Challenges by 2025" addresses an important and timely topic on the global threat of highly pathogenic avian influenza (HPAI), particularly H5N1 and H7N9. It provides a broad overview of epidemiology, molecular adaptations, zoonotic potential, and preparedness strategies. However, the paper shows a high degree of similarity with the published review by Possas et al. (Front. Public Health, July 2025) and currently lacks sufficient novelty. Several sections appear derivative in structure and content, and the manuscript reads more as a general narrative review than as a contribution offering new insights. The authors should clarify their unique contribution.
Author Response:
We thank the reviewer for this important observation. We carefully reviewed the work of Possas et al. (Front. Public Health, July 2025) and compared it with our manuscript. We agree that both reviews address highly pathogenic avian influenza (HPAI) as a global health concern, but we emphasize that our manuscript makes a distinct and complementary contribution, particularly through comparative epidemiology, molecular detail, clinical synthesis, and applied preparedness strategies.
Key Distinctions Between the Two Reviews
-
Scope of Viruses
-
Possas et al. primarily focus on H5N1, framed largely around the absence of vaccines in a high-lethality pandemic scenario.
-
Our review systematically tracks the progression of both H5N1 and H7N9, presenting a structured comparison across epidemiology, evolution, zoonotic potential, clinical outcomes, and global response (line no. 127, 232, 202, 177).
-
-
Approach and Data Presentation
-
Possas et al. 2025 present a forward-looking on emphasizing governance, AI, and vaccine preparedness.
-
Our work integrates quantitative epidemiological tables and figures (cumulative human cases, outbreak distributions, clinical features, and mutation profiles) that provide a data-driven synthesis absent in the Possas review (Line no. 127, 453).
-
-
Molecular and Clinical Insights
- Possas et al. 2025 did not discussed or present key mutations in NA and NP, highlighting mechanisms of host adaptation,
-
Our manuscript includes a detailed examination of key mutations in NA and NP, highlighting mechanisms of host adaptation, neuroinvasion, and antiviral resistance (in section 3).
-
We also compare clinical manifestations, diagnostic methods, and therapeutic responses of H5N1 vs. H7N9, offering practical insights for clinicians and policymakers ( sub section 4.3.).
-
These clinical aspects are not addressed in Possas et al. 2025
-
our introduction is grounded in a comparative, data-driven virological and epidemiological framework, with emphasis on:
-
Detailed molecular and evolutionary background (segmental genome, HA/NA subtypes, clades, and mutation-driven host adaptation);
-
Comparative epidemiological timelines of H5N1, H7N9, and H9N2, supported by 4 tables with detailed information and 1 figures.
-
Emerging One Health risks, especially novel reservoirs such as dairy cattle, which have not been highlighted in Possas et al.
-
- Conclusions — A Comparative Look
- Possas et al. conclude with an emphasis on the urgent need for vaccine innovation, AI integration, and equitable global governance, calling for a “shift toward faster action and coordinated strategy before the next pandemic begins.”
- Our conclusions focus on a broader One Health preparedness framework, recommending:
- strengthened global surveillance across birds, cattle, and humans;
- detailed genetic monitoring of mutations;
- rapid and scalable vaccine development;
- antiviral and therapeutic advancements;
- stringent farm-level and occupational biosecurity;
- structured decision-making frameworks; and
- long-term reforms in farming practices, wildlife trade, and food safety (e.g., dairy pasteurization).
Thus, while Possas et al. 2025 underscore strategic governance gaps and vaccine limitations, our review provides a comparative, multidisciplinary roadmap grounded in epidemiological data and clinical outcomes.